# Saddle-to-Saddle Dynamics
# in Diagonal Linear Networks

**Scott Pesme**
EPFL
scott.pesme@epfl.ch

**Nicolas Flammarion**
EPFL
nicolas.flammarion@epfl.ch

## Abstract

In this paper we fully describe the trajectory of gradient flow over 2-layer diagonal linear networks for the regression setting in the limit of vanishing initialisation. We show that the limiting flow successively jumps from a saddle of the training loss to another until reaching the minimum $\ell_1$-norm solution. We explicitly characterise the visited saddles as well as the jump times through a recursive algorithm reminiscent of the LARS algorithm used for computing the Lasso path. Starting from the zero vector, coordinates are successively activated until the minimum $\ell_1$-norm solution is recovered, revealing an incremental learning. Our proof leverages a convenient arc-length time-reparametrisation which enables to keep track of the transitions between the jumps. Our analysis requires negligible assumptions on the data, applies to both under and overparametrised settings and covers complex cases where there is no monotonicity of the number of active coordinates. We provide numerical experiments to support our findings.

## 1 Introduction

Strikingly simple algorithms such as gradient descent are driving forces for deep learning and have led to remarkable empirical results. Nonetheless, understanding the performances of such methods remains a challenging and exciting mystery: (i) their global convergence on highly non-convex losses is far from being trivial and (ii) the fact that they lead to solutions which generalise well [53] is still not fully understood.

To explain this second point, a major line of work has focused on the concept of implicit regularisation: amongst the infinite space of zero-loss solutions, the optimisation process must be implicitly biased towards solutions which have good generalisation properties for the considered real-world prediction tasks. Many papers have therefore shown that gradient methods have the fortunate property of asymptotically leading to solutions which have a well-behaving structure [38, 24, 16].

Aside from these results which mostly focus on characterising the asymptotic solution, a slightly different point of view has been to try to describe the full trajectory. Indeed it has been experimentally observed that gradient methods with small initialisations have the property of learning models of increasing complexity across the training of neural networks [29]. This behaviour is usually referred to as *incremental learning* or as a *saddle-to-saddle process* and describes learning curves which are piecewise constant: the training process makes very little progress for some time, followed by a sharp transition where a new "feature" is suddenly learned. In terms of optimisation trajectory, this corresponds to the iterates "jumping" from a saddle of the training loss to another.

Several settings exhibiting such dynamics for small initialisation have been considered: matrix and tensor factorisation [44, 27], simplified versions of diagonal linear networks [23, 7], linear networks [22, 45, 26], 2-layer neural networks with orthogonal inputs [10], learning leap functions with 2-layer neural networks [1] and matrix sensing [2, 33, 28]. However, all these results require

37th Conference on Neural Information Processing Systems (NeurIPS 2023).

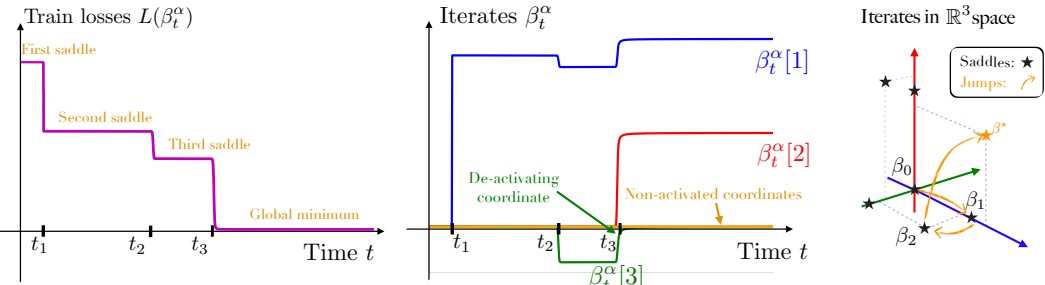

Figure 1: Gradient flow $(\beta_t^\alpha)_t$ with small initialisation scale $\alpha$ over a 2-layer diagonal linear network (for the precise experimental setting, see Appendix A). *Left:* Training loss across time, the learning is piecewise constant. *Middle:* The magnitudes of the coordinates are plotted across time: the process is piecewise constant. *Right:* In the $\mathbb{R}^3$ space in which the iterates evolve (the remaining coordinates stay at 0), the iterates jump from a saddle of the training loss to another. The jumping times $t_i$ as well as the visited saddles $\beta_i$ are entirely predicted by our theory.

restrictive assumptions on the data or only characterise the first jump. Obtaining a complete picture of the saddle-to-saddle process by describing all the visited saddles and jump times is mathematically challenging and still missing. We intend to fill this gap by considering diagonal linear networks which are simplified neural networks that have received significant attention lately [50, 48, 25, 43, 20] as they are ideal proxy models for gaining a deeper understanding of complex phenomenons such as saddle-to-saddle dynamics.

## 1.1 Informal statement of the main result

In this paper, we provide a full description of the trajectory of gradient flow over 2-layer diagonal linear networks in the limit of vanishing initialisation. The main result is informally presented here.

**Theorem 1** (Main result, informal). *In the regression setting and in the limit of vanishing initialisation, the trajectory of gradient flow over a 2-layer diagonal linear network converges towards a limiting process which is piecewise constant: the iterates successively jump from a saddle of the training loss to another, each visited saddle and jump time can recursively be computed through an algorithm (Algorithm 1) reminiscent of the LARS algorithm for the Lasso.*

The incremental learning stems from the particular structure of the saddles as they correspond to minimisers of the training loss with a constraint on the set of non-zero coordinates. The saddles therefore correspond to sparse vectors which partially fit the dataset. For simple datasets, a consequence of our main result is that **the limiting trajectory successively starts from the zero vector and successively learns the support of the sparse ground truth vector until reaching it**. **However, we make minimal assumptions on the data and our analysis also holds for complex datasets**. In that case, the successive active sets are not necessarily increasing in size and coordinates can deactivate as well as activate until reaching the minimum $\ell_1$-norm solution (see Figure 1 (middle) for an example of a deactivating coordinate). The regression setting and the diagonal network architecture are introduced in Section 2. Section 3 provides an intuitive construction of the limiting saddle-to-saddle dynamics and presents the algorithm that characterises it. Our main result regarding the convergence of the iterates towards this process is presented in Section 4 and further discussion is provided in Section 5.

## 2 Problem setup and leveraging the mirror structure

### 2.1 Setup

**Linear regression.** We study a linear regression problem with inputs $(x_1, \ldots, x_n) \in (\mathbb{R}^d)^n$ and outputs $(y_1, \ldots, y_n) \in \mathbb{R}^n$. We consider the typical quadratic loss:

$$L(\beta) = \frac{1}{2n} \sum_{i=1}^{n} (\langle \beta, x_i \rangle - y_i)^2 . \tag{1}$$

We make no assumption on the number of samples $n$ nor the dimension $d$. The only assumption we make on the data throughout the paper is that the inputs $(x_1, \ldots, x_n)$ are in *general position*. In order to state this assumption, let $X \in \mathbb{R}^{n \times d}$ be the feature matrix whose $i^{th}$ row is $x_i$ and let $\tilde{x}_j \in \mathbb{R}^n$ be its $j^{th}$ column for $j \in [d]$.

**Assumption 1** (General position). *For any $k \leq \min(n,d)$ and arbitrary signs $\sigma_1, \ldots, \sigma_k \in \{-1, 1\}$, the affine span of any $k$ points $\sigma_1 \tilde{x}_{j_1}, \ldots, \sigma_k \tilde{x}_{j_k}$ does not contain any element of the set $\{\pm \tilde{x}_j, j \neq j_1, \ldots, j_k\}$.*

This assumption is slightly technical but is standard in the Lasso literature [47]. Note that it is not restrictive as it is almost surely satisfied when the data is drawn from a continuous probability distribution [47, Lemma 4]. Letting $\mathcal{S} = \arg\min_\beta L(\beta)$ denote the affine space of solutions, Assumption 1 ensures that the minimisation problem $\min_{\beta^\star \in \mathcal{S}} \|\beta^\star\|_1$ has a unique minimiser which we denote $\beta^\star_{\ell_1}$ and which corresponds to the minimum $\ell_1$-norm solution.

**2-layer diagonal linear network.** In an effort to understand the training dynamics of neural networks, we consider a 2-layer diagonal linear network which corresponds to writing the regression vector $\beta$ as

$$\beta_w = u \odot v \ \text{ where } \ w = (u,v) \in \mathbb{R}^{2d} \,. \tag{2}$$

This parametrisation can be interpreted as a simple neural network $x \mapsto \langle u, \sigma(\mathrm{diag}(v)x) \rangle$ where $u$ are the output weights, the diagonal matrix $\mathrm{diag}(v)$ represents the inner weights, and the activation $\sigma$ is the identity function. We refer to $w = (u,v) \in \mathbb{R}^{2d}$ as the *weights* and to $\beta := u \odot v \in \mathbb{R}^d$ as the *prediction parameter*. With the parametrisation (2), the loss function $F$ over the parameters $w = (u,v) \in \mathbb{R}^{2d}$ is defined as:

$$F(w) := L(u \odot v) = \frac{1}{2n} \sum_{i=1}^{n} (\langle u \odot v, x_i \rangle - y_i)^2 \,. \tag{3}$$

Though this reparametrisation is simple, the associated optimisation problem is non-convex and highly non-trivial training dynamics already occur. The critical points of the function $F$ exhibit a very particular structure, as highlighted in the following proposition proven in Appendix B.

**Proposition 1.** *All the critical points $w_c$ of $F$ which are not global minima, i.e., $\nabla F(w_c) = \mathbf{0}$ and $F(w_c) > \min_w F(w)$, are necessarily saddle points (*i.e., *not local extrema). They map to parameters $\beta_c = u_c \odot v_c$ which satisfy $|\beta_c| \odot \nabla L(\beta_c) = \mathbf{0}$ and:*

$$\beta_c \in \underset{\beta[i]=0 \text{ for } i \notin \mathrm{supp}(\beta_c)}{\arg\min} L(\beta) \tag{4}$$

*where $\mathrm{supp}(\beta_c) = \{i \in [d], \beta_c[i] \neq 0\}$ corresponds to the support of $\beta_c$.*

The optimisation problem in Eq. (4) states that the saddle points of the train loss $F$ correspond to **sparse vectors that minimise the loss function $L$ over its non-zero coordinates**. This property already shows that the saddle points possess interesting properties from a learning perspective. In the following we loosely use the term of 'saddle' to refer to points $\beta_c \in \mathbb{R}^d$ solution of Eq. (4) **that are not saddles of the convex loss function $L$**. We adopt this terminology because they correspond to points $w_c \in \mathbb{R}^{2d}$ that are indeed saddles of the non-convex loss $F$.

**Gradient Flow and necessity of "accelerating" time.** We minimise the loss $F$ using gradient flow:

$$\mathrm{d}w_t = -\nabla F(w_t)\mathrm{d}t \,, \tag{5}$$

initialised at $u_0 = \sqrt{2}\alpha\mathbf{1} \in \mathbb{R}^d_{>0}$ with $\alpha > 0$, and $v_0 = \mathbf{0} \in \mathbb{R}^d$. This initialisation results in $\beta_0 = \mathbf{0} \in \mathbb{R}^d$ independently of the chosen weight initialisation scale $\alpha$. We denote $\beta_t^\alpha := u_t^\alpha \odot v_t^\alpha$ the prediction iterates generated from the gradient flow to highlight its dependency on the initialisation scale $\alpha$[1]. The origin $\mathbf{0} \in \mathbb{R}^{2d}$ is a critical point of the function $F$ and taking the initialisation $\alpha \to 0$ therefore arbitrarily slows down the dynamics. In fact, it can be easily shown for any fixed time $t$, that $(u_t^\alpha, v_t^\alpha) \to \mathbf{0}$ as $\alpha \to 0$, indicating that the iterates are stuck at the origin. Therefore if we restrict ourselves to a finite time analysis, there is no hope of exhibiting the observed saddle-to-saddle behaviour. To do so, we must find an appropriate bijection $\tilde{t}_\alpha$ in $\mathbb{R}_{\geq 0}$ which "accelerates" time (*i.e.* $\tilde{t}_\alpha(t) \underset{\alpha \to 0}{\longrightarrow} +\infty$ for all $t$) and consider the accelerated iterates $\beta_{\tilde{t}_\alpha(t)}^\alpha$ which can escape the saddles. Finding this bijection becomes very natural once the mirror structure is unveiled.

---

[1]We point out that the trajectory of $\beta_t^\alpha$ exactly matches that of another common parametrisation $\beta_w := \frac{1}{2}(w_+^2 - w_-^2)$, with initialisation $w_{+,0} = w_{-,0} = \alpha\mathbf{1}$.

## 2.2 Leveraging the mirror flow structure

While the iterates $(w_t^\alpha)_t$ follow a gradient flow on the non-convex loss $F$, it is shown in [5] that the iterates $\beta_t^\alpha$ follow a mirror flow on the convex loss $L$ with potential $\phi_\alpha$ and initialisation $\beta_{t=0}^\alpha = \mathbf{0}$:

$$d\nabla\phi_\alpha(\beta_t^\alpha) = -\nabla L(\beta_t^\alpha)dt, \tag{6}$$

where $\phi_\alpha$ is the hyperbolic entropy function [21] defined as:

$$\phi_\alpha(\beta) = \frac{1}{2}\sum_{i=1}^{d}\left(\beta_i\text{arcsinh}(\frac{\beta_i}{\alpha^2}) - \sqrt{\beta_i^2 + \alpha^4} + \alpha^2\right). \tag{7}$$

Unveiling the mirror flow structure enables to leverage convex optimisation tools to prove convergence of the iterates to a global minimiser $\beta_\alpha^\star$ as well as a simple proof of the implicit regularisation problem it solves. As shown by Woodworth et al. [50], in the overparametrised setting where $d > n$ and where there exists an infinite number of global minima, the limit $\beta_\alpha^\star$ is the solution of the problem:

$$\beta_\alpha^\star = \underset{y_i=\langle x_i,\beta\rangle,\forall i}{\arg\min}\ \phi_\alpha(\beta). \tag{8}$$

Furthermore, a simple function analysis shows that $\phi_\alpha$ behaves as a rescaled $\ell_1$-norm as $\alpha$ goes to 0, meaning that the recovered solution $\beta_\alpha^\star$ converges to the minimum $\ell_1$-norm solution $\beta_{\ell_1}^\star = \arg\min_{y_i=\langle x_i,\beta\rangle}\|\beta\|_1$ as $\alpha$ goes to 0 (see [49] for a precise rate). To bring to light the saddle-to-saddle dynamics which occurs as we take the initialisation to 0, we make substantial use of the nice mirror structure from Eq. (6).

**Appropriate time rescaling.** To understand the limiting dynamics of $\beta_t^\alpha$, it is natural to consider the limit $\alpha \to 0$ in Eq. (6). However, the potential $\phi_\alpha$ is such that $\phi_\alpha(\beta) \sim \ln(1/\alpha)\|\beta\|_1$ for small $\alpha$ and therefore degenerates as $\alpha \to 0$. Similarly, for $\beta \neq \mathbf{0}$, $\|\nabla\phi_\alpha(\beta)\| \to \infty$ as $\alpha \to 0$. The formulation from Eq. (6) is thus not appropriate to take the limit $\alpha \to 0$. We can nonetheless obtain a meaningful limit by considering the opportune time acceleration $\tilde{t}_\alpha(t) = \ln(1/\alpha) \cdot t$ and looking at the accelerated iterates

$$\tilde{\beta}_t^\alpha := \beta_{\tilde{t}_\alpha(t)}^\alpha = \beta_{\ln(1/\alpha)t}^\alpha. \tag{9}$$

Indeed, a simple chain rule leads to the "accelerated mirror flow": $d\nabla\phi_\alpha(\tilde{\beta}_t^\alpha) = -\ln\left(\frac{1}{\alpha}\right)\nabla L(\tilde{\beta}_t^\alpha)dt$. The accelerated iterates $(\tilde{\beta}_t^\alpha)_t$ follow a mirror descent with a rescaled potential:

$$d\nabla\tilde{\phi}_\alpha(\tilde{\beta}_t^\alpha) = -\nabla L(\tilde{\beta}_t^\alpha)dt, \qquad \text{where} \qquad \tilde{\phi}_\alpha := \frac{1}{\ln(1/\alpha)} \cdot \phi_\alpha, \tag{10}$$

with $\tilde{\beta}_{t=0} = \mathbf{0}$ and where $\phi_\alpha$ is defined Eq. (7). Our choice of time acceleration ensures that the rescaled potential $\tilde{\phi}_\alpha$ is non-degenerate as the initialisation goes to 0 since $\tilde{\phi}_\alpha(\beta) \underset{\alpha\to0}{\sim} \|\beta\|_1$.

## 3 Intuitive construction of the limiting flow and saddle-to-saddle algorithm

In this section, we aim to give a comprehensible construction of the limiting flow. We therefore choose to provide intuition over pure rigor, and defer the full and rigorous proof to the Appendix E. The technical crux of our analysis is to demonstrate the existence of a piecewise constant limiting process towards which the iterates $\tilde{\beta}^\alpha$ converge to. The convergence result is deferred to the following Section 4. **In this section we assume this convergence and refer to this piecewise constant limiting process as $(\tilde{\beta}_t^\circ)_t$.** Our goal is then to determine the jump times $(t_1, \ldots, t_p)$ as well as the saddles $(\beta_0, \ldots, \beta_p)$ which fully define this process.

To do so, it is natural to examine the limiting equation obtained when taking the limit $\alpha \to 0$ in Eq. (10). We first turn to its integral form which writes:

$$-\int_0^t \nabla L(\tilde{\beta}_s^\alpha)ds = \nabla\tilde{\phi}_\alpha(\tilde{\beta}_t^\alpha). \tag{11}$$

Provided the convergence of the flow $\tilde{\beta}^\alpha$ towards $\tilde{\beta}^\circ$, the left hand side of the previous equation converges to $-\int_0^t \nabla L(\tilde{\beta}_s^\circ)\mathrm{d}s$. For the right hand side, recall that $\tilde{\phi}_\alpha(\beta) \overset{\alpha \to 0}{\sim} \|\beta\|_1$, it is therefore natural to expect the right hand side of Eq. (11) to converge towards an element of $\partial\|\tilde{\beta}_t^\circ\|_1$, where we recall the definition of the subderivative of the $\ell_1$-norm as:

$$\partial\|\tilde{\beta}\|_1 = \{1\} \text{ if } \tilde{\beta} > 0, \quad \{-1\} \text{ if } \tilde{\beta} < 0, \quad [-1,1] \text{ if } \tilde{\beta} = 0.$$

The arising key equation which must satisfy the limiting process $\tilde{\beta}^\circ$ is then, for all $t \geq 0$:

$$-\int_0^t \nabla L(\tilde{\beta}_s^\circ)\mathrm{d}s \in \partial\|\tilde{\beta}_t^\circ\|_1. \tag{12}$$

We show that **this equation uniquely determines the piecewise constant process** $\tilde{\beta}^\circ$ by imposing the number of jumps $p$, the jump times as well as the saddles which are visited between the jumps. Indeed the relation described in Eq. (12) provides 4 restrictive properties that enable to construct $\tilde{\beta}^\circ$. To state them, let $s_t = -\int_0^t \nabla L(\tilde{\beta}_s^\circ)\mathrm{d}s$ and notice that it is continuous and piecewise linear since $\tilde{\beta}^\circ$ is piecewise constant. For each coordinate $i \in [d]$, it holds that:

**(K1)** $s_t[i] \in [-1,1]$       **(K2)** $s_t[i] = 1 \Rightarrow \tilde{\beta}_t^\circ[i] \geq 0$ and $s_t[i] = -1 \Rightarrow \tilde{\beta}_t^\circ[i] \leq 0$

**(K3)** $s_t[i] \in (-1,1) \Rightarrow \tilde{\beta}_t^\circ[i] = 0$      **(K4)** $\tilde{\beta}_t^\circ[i] > 0 \Rightarrow s_t[i] = 1$ and $\tilde{\beta}_t^\circ[i] < 0 \Rightarrow s_t[i] = -1$

To understand how these conditions lead to the algorithm which determines the jump times and the visited saddles, we present a 2-dimensional example for which we can walk through each step. The general case then naturally follows from this simple example.

### 3.1 Construction of the saddle-to-saddle algorithm with an illustrative $2d$ example.

Let us consider $n = d = 2$ and data matrix $X \in \mathbb{R}^{2 \times 2}$ such that $X^\top X = ((1, 0.2), (0.2, -0.2))$. We consider $\beta^\star = (-0.2, 2) \in \mathbb{R}^2$ and outputs $y = X\beta^\star$. This setting is such that the loss $L$ has $\beta^\star$ as its unique minimum and $L(\beta^*) = 0$. Furthermore the non-convex loss $F$ has 3 saddles which map to: $\beta_{c,0} := (0,0) = \arg\min_{\beta_i = 0, \forall i} L(\beta)$, $\beta_{c,1} := (0.2, 0) = \arg\min_{\beta[2]=0} L(\beta)$ and $\beta_{c,2} := (0, 1.6) = \arg\min_{\beta[1]=0} L(\beta)$. The loss function $L$ is sketched in Figure 2 (*Left*). Notice that by the definition of $\beta_{c,1}$ and $\beta_{c,2}$, the gradients of the loss at these points are orthogonal to the axis they belong to. When running gradient flow with a small initialisation over our diagonal linear network, we obtain the plots illustrated Figure 2 (*Middle and Right*). We observe three jumps: the iterates jump from the saddle at the origin to $\beta_{c,1}$ at time $t_1$, then to $\beta_{c,2}$ at time $t_2$ and finally to the global minimum $\beta^\star$ at time $t_3$.

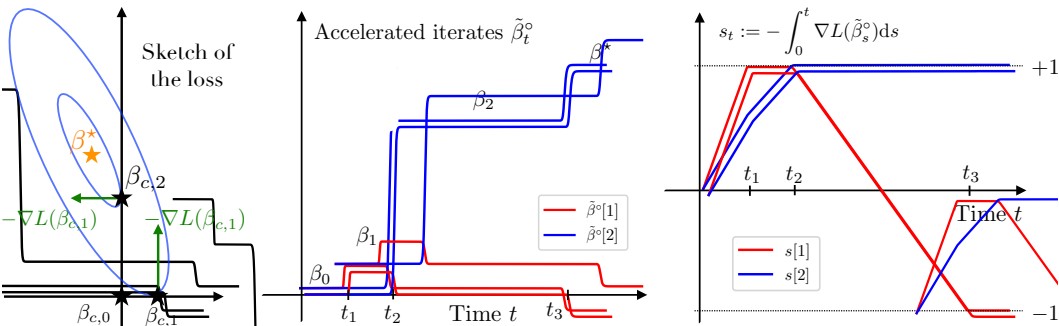

Figure 2: *Left*: Sketch of the $2d$ loss. *Middle and right*: Outputs of gradient flow with small initialisation scale: the iterates are piecewise constant and $s_t$ is piecewise linear across time. We refer to the main text for further details.

Let us show how Eq. (12) enables us to theoretically recover this trajectory. A simple observation which we will use several times below is that for any $t' > t$ such that $\tilde{\beta}^\circ$ is constant equal to $\beta$ over the time interval $(t, t')$, the definition of $s$ enables to write that $s_{t'} = s_t - (t' - t) \cdot \nabla L(\beta)$.

**Zeroth saddle:** The iterates are at the saddle at the origin: $\tilde{\beta}_t^\circ = \beta_0 := \beta_{c,0}$ and therefore $s_t = -t \cdot \nabla L(\beta_0)$. Our key equation Eq. (12) is verified since $s_t = -t \cdot \nabla L(\beta_0) \in \partial \|\beta_0\|_1 = [-1, 1]^d$. However the iterates cannot stay at the origin after time $t_1 := 1/\|\nabla L(\beta_0)\|_\infty$ which corresponds to the time at which the first coordinate of $s_t$ hits $+1$: $s_{t_1}[1] = 1$. If the iterates stayed at the origin after $t_1$, (K1) for $i = 1$ would be violated. The iterates must hence jump.

**First saddle:** The iterates can only jump to a point different from the origin which maintains Eq. (12) valid. We denote this point as $\beta_1$. Notice that:

- $s_{t_1}[2] = -t_1 \cdot \nabla L(\beta_0)[2] \in (-1, 1)$ and since $s_t$ is continuous, we must have $\beta_1[2] = 0$ (K3)
- $s_{t_1}[1] = 1$ and hence for $t \geq t_1$, $s_t[1] = 1 - (t - t_1)\nabla L(\beta_1)[1]$. We cannot have $\nabla L(\beta_1)[1] < 0$ (K1), and neither $\nabla L(\beta_1)[1] > 0$ since otherwise $s_t[1] \in (-1, 1)$ and $\beta_1 = \mathbf{0}$ (K3)

The two conditions $\beta_1[2] = 0$ and $\nabla L(\beta_1)[1] = 0$ **uniquely defines** $\beta_1$ **as equal to** $\beta_{c,1}$. We now want to know if and when the iterates jump again. We saw that $s_t[1]$ remains at the value $+1$. However since $\beta_1$ is not a global minimum, $\nabla L(\beta_1)[2] \neq 0$ and $s_t[2]$ hits $+1$ at time $t_2$ defined such that $-(t_1 \nabla L(\beta_0) + (t_2 - t_1)\nabla L(\beta_1))[2] = 1$. The iterates must jump otherwise (K1) would break.

**The iterates cannot jump to $\beta^\star$ yet!** As the second coordinate of the iterates can activate, one could expect the iterates to be able to jump to the global minimum. However note that $s_t$ is a continuous function and that $s_{t_2}$ is equal to the vector $(1, 1)$. If the iterates jumped to the global minimum, then the first coordinate of the iterates would change sign from $+0.2$ to $-0.2$. Due to (K4) this would lead $s_t$ jumping from $+1$ to $-1$, violating its continuity.

**Second saddle:** We denote as $\beta_2$ the point to which the iterates jump. $s_{t_2}$ is now equal to the vector $(1, 1)$ and therefore *(i)* $\beta_2 \geq 0$ (coordinate-wise) from (K2 and K3) and the continuity of $s$. Since $s_t = s_{t_2} - (t - t_2)\nabla L(\beta_2)$, we must also have: *(ii)* $\nabla L(\beta_2) \geq 0$ from (K1) *(iii)* for $i \in \{1, 2\}$, if $\beta_2[i] \neq 0$ then $\nabla L(\beta_2)[i] = 0$ from (K4). The three conditions *(i)*, *(ii)* and *(iii)* precisely correspond to the optimality conditions of the following problem:

$$\underset{\beta[1] \geq 0, \beta[2] \geq 0}{\arg \min} \ L(\beta).$$

The unique minimiser of this problem is $\beta_{c,2}$, hence $\beta_2 = \beta_{c,2}$, which means that the first coordinate deactivates. Similar to before, (K1) is valid until the time $t_3$ at which the first coordinate of $s_t = s_{t_2} - (t - t_2)\nabla L(\beta_2)$ reaches $-1$ due to the fact that $\nabla L(\beta_2)[1] > 0$.

**Global minimum:** We follow the exact same reasoning as for the second saddle. We now have $s_{t_3}$ equal to the vector $(-1, 1)$ and the iterates must jump to a point $\beta_3$ such that *(i)* $\beta_3[1] \leq 0$, $\beta_3[2] \geq 0$ (K2 and K3), *(ii)* $\nabla L(\beta_3)[1] \leq 0$, $\nabla L(\beta_3)[2] \geq 0$ (K1), *(iii)* for $i \in \{1, 2\}$, if $\beta_3[i] \neq 0$ then $\nabla L(\beta_3)[i] = 0$ (K4). Again, these are the optimality conditions of the following problem:

$$\underset{\beta[1] \leq 0, \beta[2] \geq 0}{\arg \min} \ L(\beta).$$

$\beta^\star$ is the unique minimiser of this problem and $\beta_3 = \beta^\star$. For $t \geq t_3$ we have $s_t = s_{t_3}$ and Eq. (12) is satisfied for all following times: the iterates do not have to move anymore.

## 3.2 Presentation of the full saddle-to-saddle algorithm

We can now provide the full algorithm (Algorithm 1) which computes the jump times $(t_1, \ldots, t_p)$ and saddles $(\beta_0 = \mathbf{0}, \beta_1, \ldots, \beta_p)$ as the values and vectors such that the associated piecewise constant process satisfies Eq. (12) for all $t$. This algorithm therefore defines our limiting process $\tilde{\beta}^\circ$.

**Algorithm 1 in words.** The algorithm is a concise representation of the steps we followed in the previous section to construct $\tilde{\beta}^\circ$. We explain each step in words below. Starting from $k = 0$, assume we enter the loop number $k$ at the saddle $\beta_k$ computed in the previous loop:

- The set $\mathcal{A}_k$ contains the set of coordinates "which are unstable": by having a non-zero derivative, the loss could be decreased by moving along each one of these coordinates and one of these coordinates will have to activate.

**Algorithm 1:** Successive saddles and jump times of $\lim_{\alpha \to 0} \tilde{\beta}^\alpha$

---

**Initialise:** $(t, \beta, s) \leftarrow (0, \mathbf{0}, \mathbf{0})$;

**while** $\nabla L(\beta) \neq \mathbf{0}$ **do**

$\quad \mathcal{A} \leftarrow \{j \in [d], \nabla L(\beta)(j) \neq 0\}$

$\quad \Delta \leftarrow \inf \{\delta > 0 \text{ s.t. } \exists i \in \mathcal{A}, \ s(i) - \delta \nabla L(\beta)(i) = \pm 1\}$

$\quad (t, \ s) \leftarrow (t + \Delta, \ s - \Delta \cdot \nabla L(\beta))$

$\quad \beta \leftarrow \arg\min \ L(\beta) \text{ where } \beta \in \left\{\beta \in \mathbb{R}^d \text{ s.t. } \begin{array}{l} \beta_i \geq 0 \text{ if } s(i) = +1 \\ \beta_i \leq 0 \text{ if } s(i) = -1 \\ \beta_i = 0 \text{ if } s(i) \in (-1,1) \end{array}\right\}$

**end**

**Output:** Successive values of $\beta$ and $t$

---

- The time gap $\Delta_k$ corresponds to the time spent at the saddle $\beta_k$. It is computed as being the elapsed time just before (K1) breaks if the coordinates do not jump.

- We update $t_{k+1} = t_k + \Delta_k$ and $s_{k+1} = s_k - \Delta_k \nabla L(\beta_k)$: $t_{k+1}$ corresponds to the time at which the iterates leave the saddle $\beta_k$ and $s_{k+1}$ constrains the signs of the next saddle $\beta_{k+1}$

- The solution $\beta_{k+1}$ of the constrained minimisation problem is the saddle to which the flow jumps to at time $t_{k+1}$. The optimality conditions of this problem are such that Eq. (12) is maintained for $t \geq t_{k+1}$.

**Various comments on Algorithm 1.** First we point out that any solution $\beta_c$ of the constrained minimisation problem which appears in Algorithm 1 also satisfies $\beta_c = \arg\min_{\beta[i]=0 \text{ for } i \notin \text{supp}(\beta_c)} L(\beta)$ as in Eq. (4): the algorithm hence indeed outputs saddles as expected. Up until now we have never checked whether the algorithm's constrained minimisation problem has a unique minimum. This is crucial otherwise the assignment step would be ill-defined. Showing the uniqueness is non-trivial and is guaranteed thanks to the general position Assumption 1 on the data (see Proposition 7 in Appendix D.1). In this same proposition, we also show that the algorithm terminates in at most $\min\left(2^d, \sum_{k=0}^n \binom{d}{k}\right)$ steps, that the loss strictly decreases at each step and that the final output $\beta_p$ is the minimum $\ell_1$-norm solution. These last two properties are expected given the fact that the algorithm arises as being the limit process of $\tilde{\beta}^\alpha$ which follows the mirror flow Eq. (10).

**Links with the LARS algorithm for the Lasso.** Recall that the Lasso problem [46, 15] is formulated as:

$$\beta_\lambda^\star = \underset{\beta \in \mathbb{R}^d}{\arg\min} \ L(\beta) + \lambda\|\beta\|_1. \tag{13}$$

The optimality condition of Eq. (13) writes $-\nabla L(\beta_\lambda^\star) \in \lambda\partial\|\beta_\lambda^\star\|_1$. Now notice the similarity with Eq. (12): the two would be equivalent with $\lambda = 1/t$ if the integration on the left hand side of Eq. (12) did not average over the whole trajectory but only on the final iterate, in which case $-\int_0^t \nabla L(\tilde{\beta}_t^\circ)\mathrm{d}s = -t \cdot \nabla L(\tilde{\beta}_t^\circ)$. Though the difference is small, the trajectories of our limiting trajectory $\tilde{\beta}^\circ$ and the lasso path $(\beta_\lambda^\star)_\lambda$ are quite different: one has jumps, whereas the other is continuous. Nonetheless, the construction of Algorithm 1 shares many similarities with that of the Least Angle Regression (LARS) algorithm [19] (originally named the Homotopy algorithm [39]) which is used to compute the Lasso path. A notable difference however is the fact that each step of our algorithm depends on the whole trajectory through the vector $s$, whereas the LARS algorithm can be started from any point on the path.

### 3.3 Outputs of the algorithm under a RIP and gap assumption on the data.

Unlike previous results on incremental learning, complex behaviours can occur when the feature matrix is ill designed: several coordinates can activate and deactivate at the same time (see Appendix A for various cases). However, if the feature matrix satisfies the $2r$-restricted isometry property (RIP) [14] and there exists an $r$-sparse solution $\beta^\star$, the visited saddles can be easily approximated using Algorithm 1. We provide the precise characterisation below.

**Sparse regression with RIP and gap assumption.** *(RIP) Assume that there exists an $r$-sparse vector $\beta^\star$ such that $y_i = \langle x_i, \beta^\star \rangle$. Furthermore we assume that the feature matrix $X \in \mathbb{R}^{n,d}$ satisfies the $2r$-restricted isometry property with constant $\tilde{\varepsilon} < \sqrt{2} - 1 < 1/2$: i.e. for all submatrix $X_s$ where we extract any $s \leq 2r$ columns of $X$, the matrix $X_s^\top X_s / n$ of size $s \times s$ has all its eigenvalues in the interval $[1 - \tilde{\varepsilon}, 1 + \tilde{\varepsilon}]$. (Gap assumption) Furthermore we assume that the $r$-sparse vector $\beta^\star$ has coordinates which have a "sufficient gap'. W.l.o.g we write $\beta^\star = (\beta_1^\star, \ldots, \beta_r^\star, 0, \ldots, 0)$ with $|\beta_1^\star| \geq \ldots \geq |\beta_r^\star| > 0$ and we define $\lambda := \min_{i \in [r]}(|\beta_i^\star| - |\beta_{i+1}^\star|) \geq 0$ which corresponds to the smallest gap between the entries of $|\beta^\star|$. We assume that $5\tilde{\varepsilon}\|\beta^\star\|_2 < \lambda/2$ and we let $\varepsilon := 5\tilde{\varepsilon}$.*

A classic result from compressed sensing (see Candes [13, Theorem 1.2]) is that the $2r$-restricted isometry property with constant $\sqrt{2} - 1$ ensures that the minimum $\ell_0$-minimisation problem has a unique $r$-sparse solution which is $\beta^\star$. This means that Algorithm 1 will have $\beta^\star$ as final output and the following proposition shows that we can precisely characterise each of its outputs when the data satisfies the previous assumptions.

**Proposition 2.** *Under the restricted isometry property and the gap assumption stated right above, Algorithm 1 terminates in $r$-loops and outputs:*

$$\beta_1 = (\beta_1[1], 0, \ldots, 0) \qquad \text{with} \qquad \beta_1[1] \in [\beta_1^\star - \varepsilon\|\beta^\star\|, \beta_2^\star + \varepsilon\|\beta^\star\|]$$

$$\beta_2 = (\beta_2[1], \beta_2[2], 0, \ldots, 0) \qquad \text{with} \quad \begin{cases} \beta_2[1] \in [\beta_1^\star - \varepsilon\|\beta^\star\|, \beta_1^\star + \varepsilon\|\beta^\star\|] \\ \beta_2[2] \in [\beta_2^\star - \varepsilon\|\beta^\star\|, \beta_2^\star + \varepsilon\|\beta^\star\|] \end{cases}$$

$$\vdots$$

$$\beta_{r-1} = (\beta_{r-1}[1], \ldots, \beta_{r-1}[r-1], 0, \ldots, 0) \quad \text{with } \beta_{r-1}[i] \in [\beta_i^\star - \varepsilon\|\beta^\star\|, \beta_i^\star + \varepsilon\|\beta^\star\|]$$
$$\beta_r = \beta^\star = (\beta_1^\star, \ldots, \beta_r^\star, 0, \ldots, 0),$$

*at times $t_1, \ldots, t_r$ such that $t_i \in \left[ \frac{1}{|\beta_i^\star| + \varepsilon\|\beta^\star\|}, \frac{1}{|\beta_i^\star| - \varepsilon\|\beta^\star\|} \right]$ and where $\|\cdot\|$ denotes the $\ell_2$ norm.*

Informally, this means that the algorithm terminates in exactly $r$ loops and outputs jump times and saddles roughly equal to $t_i = 1/|\beta_i^\star|$ and $\beta_i = (\beta_1^\star, \cdots, \beta_i^\star, 0, \ldots, 0)$. Therefore, in simple settings, the support of the sparse vector is learnt a coordinate at a time, without any deactivations. We refer to Appendix D.2 for the proof.

## 4 Convergence of the iterates towards the process defined by Algorithm 1

We are now fully equipped to state our main result which formalises the convergence of the accelerated iterates towards the limiting process $\tilde{\beta}^\circ$ which we built in the previous section.

**Theorem 2.** *Let the saddles $(\beta_0 = \mathbf{0}, \beta_1, \ldots, \beta_{p-1}, \beta_p = \beta_{\ell_1}^\star)$ and jump times $(t_0 = 0, t_1, \ldots, t_p)$ be the outputs of Algorithm 1 and let $(\tilde{\beta}_t^\circ)_t$ be the piecewise constant process defined as follows:*

$$\textbf{(Saddles)} \qquad \tilde{\beta}_t^\circ = \beta_k \qquad \text{for } t \in (t_k, t_{k+1}) \text{ and } 0 \leq k \leq p, \ t_{p+1} = +\infty.$$

*The accelerated flow $(\tilde{\beta}_t^\alpha)_t$ defined in Eq. (9) uniformly converges towards the limiting process $(\tilde{\beta}_t^\circ)_t$ on any compact subset of $\mathbb{R}_{\geq 0} \setminus \{t_1, \ldots, t_p\}$.*

**Convergence result.** We recall that from a technical point of view, showing the existence of a limiting process $\lim_{\alpha \to 0} \tilde{\beta}^\alpha$ is the toughest part. Theorem 2 provides this existence as well as the uniform convergence of the accelerated iterates towards $\tilde{\beta}^\circ$ over all closed intervals of $\mathbb{R}$ which do not contain the jump times. We highlight that this is the strongest type of convergence we could expect and a uniform convergence over all intervals of the form $[0, T]$ is impossible given that the limiting process $\tilde{\beta}^\circ$ is discontinuous. In Proposition 3, we give an even stronger result by showing a graph convergence of the iterates which takes into account the path followed between the jumps. We also point out that we can easily show the same type of convergence for the accelerated weights $\tilde{w}_t^\alpha := w_{\tilde{t}^\alpha(t)}^\alpha$. Indeed, using the bijective mapping which links the weights $w_t$ and the predictors $\beta_t$ (see Lemma 1 in Appendix C), we immediately get that the accelerated weights $(\tilde{u}^\alpha, \tilde{v}^\alpha)$ uniformly converge towards the limiting process $(\sqrt{|\tilde{\beta}^\circ|}, \text{sign}(\tilde{\beta}^\circ)\sqrt{|\tilde{\beta}^\circ|})$ on any compact subset of $\mathbb{R}_{\geq 0} \setminus \{t_1, \ldots, t_p\}$.

**Estimates for the non-accelerated iterates** $\beta_t^\alpha$**.** We point out that our result provides no speed of convergence of $\tilde{\beta}^\alpha$ towards $\tilde{\beta}^\circ$. We believe that a non-asymptotic result is challenging and leave it as future work. Note that we experimentally notice that the convergence rate quickly degrades after each saddle. Nonetheless, we can still write for the non-accelerated iterates that $\beta_t^\alpha = \tilde{\beta}_{t/\ln(1/\alpha)}^\alpha \sim \tilde{\beta}_{t/\ln(1/\alpha)}^\circ$ as $\alpha \to 0$. Hence, for $\alpha$ small enough the iterates $\beta_t^\alpha$ are roughly equal to 0 until time $t_1 \cdot \ln(1/\alpha)$ and the minimum $\ell_1$-norm interpolator is reached at time $t_p \cdot \ln(1/\alpha)$. **Such a precise estimate of the global convergence time is rather remarkable** and goes beyond classical Lyapunov analysises which only leads to $L(\beta_t^\alpha) \lesssim \ln(1/\alpha)/t$ (see Proposition 4 in Appendix C).

**Natural extensions of our setting.** More general initialisations can easily be dealt with. For instance, initialisations of the form $u_{t=0} = \alpha \mathbf{u_0} \in \mathbb{R}^d$ lead to the exact same result as it is shown in [50] (Discussion after Theorem 1) that the associated mirror still converges to the $\ell_1$-norm. Initialisations of the form $[u_{t=0}]_i = \alpha^{k_i}$, where $k_i > 0$, lead to the associated potential converging towards a weighted $\ell_1$-norm and one should modify Algorithm 1 by accordingly weighting $\nabla L(\beta)$ in the algorithm. Also, deeper linear architectures of the form $\beta_w = w_+^D - w_-^D$ as in [50] do not change our result as the associated mirror still converges towards the $\ell_1$-norm. Though we only consider the square loss in the paper, we believe that all our results should hold for any loss of the type $L(\beta) = \sum_{i=1}^n \ell(y_i, \langle x_i, \beta \rangle)$ where for all $y \in \mathbb{R}$, $\ell(y, \cdot)$ is strictly convex with a unique minimiser at $y$. In fact, the only property which cannot directly be adapted from our results is showing the uniform boundedness of the iterates (see discussion before Proposition 5 in Appendix C).

## 4.1 High level sketch of proof of $\tilde{\beta}^\alpha \to \tilde{\beta}^\circ$ which leverages an arc-length parametrisation

In this section, we give the high level ideas concerning the proof of the convergence $\tilde{\beta}^\alpha \to \tilde{\beta}^\circ$ given in Theorem 2. A full and detailed proof can be found in Appendix E. The main difficulty stems from the non-continuity of the limit process $\tilde{\beta}^\circ$. To circumvent this difficulty, a clever trick which we borrow to [18, 36] is to "slow-down" time when the jumps occur by considering **an arc-length parametrisation of the path**. We consider the $\mathbb{R}_{\geq 0}$ arclength bijection $\tau^\alpha$ and leverage it to define the 'appropriately slowed down' iterates $\hat{\beta}_\tau^\alpha$ as:

$$\hat{\beta}_\tau^\alpha = \tilde{\beta}_{\hat{t}^\alpha(\tau)}^\alpha \qquad \text{where} \qquad \hat{t}_\tau^\alpha = (\tau^\alpha)^{-1}(\tau) \ \text{ and } \ \tau^\alpha(t) = t + \int_0^t \|\dot{\tilde{\beta}}_s^\alpha\| \mathrm{d}s.$$

This time reparametrisation has the fortunate but crucial property of leading to $\dot{\hat{t}}^\alpha(\tau) + \|\dot{\hat{\beta}}_\tau^\alpha\| = 1$ by a simple chain rule, which means that the speed of $(\hat{\beta}_\tau^\alpha)_\tau$ **is uniformly upperbounded by** 1 **independently of** $\alpha$. This behaviour is in stark contrast with the process $(\tilde{\beta}_t^\alpha)_t$ which has a speed which explodes at the jumps. This change of time now allows us to use Arzelà-Ascoli's theorem to extract a subsequence which uniformly converges to a limiting process which we denote $\hat{\beta}$. Importantly, $\hat{\beta}$ enables to keep track of the path followed between the jumps as we show that its trajectory has two regimes:

**Saddles:** $\hat{\beta}_\tau = \beta_k$ $\qquad$ **Connections:** $\dot{\hat{\beta}}_\tau = -\dfrac{|\hat{\beta}_\tau| \odot \nabla L(\hat{\beta}_\tau)}{\||\hat{\beta}_\tau| \odot \nabla L(\hat{\beta}_\tau)\|}.$

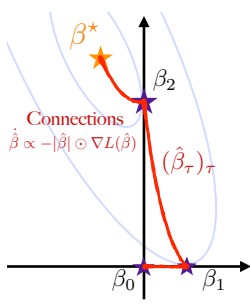

The process $\hat{\beta}$ is illustrated on the right: the red curves correspond to the paths which the iterates follow during the jumps. These paths are called *heteroclinic orbits* in the dynamical systems literature [31, 3]. To prove Theorem 2, we can map back the convergence of $\hat{\beta}^\alpha$ to show that of $\tilde{\beta}^\alpha$. Moreover from the convergence $\hat{\beta}^\alpha \to \hat{\beta}$ we get a more complete picture of the limiting dynamics of $\tilde{\beta}^\alpha$ as it naturally implies the convergence of the graph of the iterates $(\tilde{\beta}_t^\alpha)_t$ converges towards that of $(\hat{\beta}_\tau)_\tau$. The graph convergence result is formalised in this last proposition.

**Proposition 3.** *For all* $T > t_p$*, the graph of the iterates* $(\tilde{\beta}_t^\alpha)_{t \leq T}$ *converges to that of* $(\hat{\beta}_\tau)_\tau$ *:*

$$\mathrm{dist}(\{\tilde{\beta}_t^\alpha\}_{t \leq T}, \{\hat{\beta}_\tau\}_{\tau \geq 0}) \xrightarrow[\alpha \to 0]{} 0 \qquad \text{(Hausdorff distance)}$$

# 5 Further discussion and conclusion

**Link between incremental learning and saddle-to-saddle dynamics.** The incremental learning phenomenon and the saddle-to-saddle process are often complementary facets of the same idea and refer to the same phenomenon. Indeed for gradient flows $\mathrm{d}w_t = -\nabla F(w_t)\mathrm{d}t$, fixed points of the dynamics correspond to critical points of the loss. Stages with little progress in learning and minimal movement of the iterates necessarily correspond to the iterates being in the vicinity of a critical point of the loss. It turns out that in many settings (linear networks [30], matrix sensing [8, 41]), critical points are necessarily saddle points of the loss (if not global minima) and that they have a very particular structure (high sparsity, low rank, etc.). We finally note that an alternative approach to realising saddle-to-saddle dynamics is through the perturbation of the gradient flow by a vanishing noise as studied in [6].

**Characterisation of the visited saddles.** A common belief is that the saddle-to-saddle trajectory can be found by successively computing the direction of most negative curvature of the loss (i.e. the eigenvector corresponding to the most negative eigenvalue) and following this direction until reaching the next saddle [26]. However this statement cannot be accurate as it is inconsistent with our algorithm in our setting. In fact, it can be shown that this algorithm would match the orthogonal matching pursuit (OMP) algorithm [42, 17] which does not necessarily lead to the minimum $\ell_1$-norm interpolator. In [7], which is the closest to our work and the first to prove convergence of the iterates towards a piece-wise constant process, the successive saddles are entirely characterised and connected to the Lasso regularisation path in the underparameterised setting. Recently, [9] extended the diagonal linear network setting to diagonal parametrisations of the form $f_{u \odot v}$, but at the cost of stronger assumptions on the trajectory.

**Adaptive Inverse Scale Space Method.** Following the submission of our paper, we were informed that Algorithm 1 had already been proposed and analysed in the compressed sensing literature. Indeed it exactly corresponds to the Adaptive Inverse Scale Space Method (aISS) proposed in [11]. The motivations behind its study are extremely different from ours and originate from the study of Bregman iteration [12, 40, 52] which is an efficient method for solving $\ell_1$ related minimisation problems. The so-called inverse scale space flow which corresponds to Eq. (12) in our paper can be seen as the continuous version of Bregman iteration. As in our paper, [11] show that this equation can be solved through an iterative algorithm. We refer to [51, Section 2] for further details. However we did not find any results in this literature concerning the uniqueness of the constrained minimisation problem due to Assumption 1, nor on the maximum number of iterations, the behaviour under RIP assumptions and the maximum number of active coordinates.

**Subdifferential equations and rate-independent systems.** As in Eq. (12), subdifferential inclusions of the form $\nabla L(\beta_t) \in \frac{\mathrm{d}}{\mathrm{d}t}\partial h(\beta_t)$ for non-differential functions $h$ have been studied by Attouch et al. [4] but for strongly convex functions $h$. In this case, the solutions are continuous and do not exhibit jumps. On another hand, [18, 36, 37] consider so-called *rate-independent systems* of the form $\partial_q E(t, q_t) \in \partial h(\dot{q}_t)$ for 1-homogeneous *dissipation* potentials $h$. Examples of such systems are ubiquitous in mechanics and appear in problems related to friction, crack propagation, elastoplasticity and ferromagnetism to name a few [35, Ch. 6 for a survey]. As in our case, the main difficulty with such processes is the possible appearance of jumps when the energy $E$ is non-convex.

**Conclusion.** Our study examines the behaviour of gradient flow with vanishing initialisation over diagonal linear networks. We prove that it leads to the flow jumping from a saddle point of the loss to another. Our analysis characterises each visited saddle point as well as the jumping times through an algorithm which is reminiscent of the LARS method used in the Lasso framework. There are several avenues for further exploration. The most compelling one is the extension of these techniques to broader contexts for which the implicit bias of gradient flow has not yet fully been understood.

**Acknowledgments.** S.P. would like to thank Loucas Pillaud-Vivien for introducing him to this beautiful topic and for the many insightful discussions. S.P. also thanks Quentin Rebjock for the many helpful discussions and Johan S. Wind for reaching out and providing the reference of [11]. The authors also thank Jérôme Bolte for the discussions concerning subdifferential equations, Aris Daniilidis for the reference of [32], as well as Aditya Varre and Mathieu Even for proofreading the paper.

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

## A   Experimental setup and additional: experiments, extension, related works.

**Experimental setup and additional experiments.** For each experiment we generate our dataset as $y_i = \langle x_i, \beta^\star \rangle$ where $x_i = \mathcal{N}(\mathbf{0}, H)$ for a a diagonal covariance matrix $H$ and $\beta^\star$ is a vector of $\mathbb{R}^d$. Gradient descent is run with a small step size and from initialisation $u_{t=0} = \sqrt{2}\alpha\mathbf{1} \in \mathbb{R}^d$ and $v_{t=0} = \mathbf{0}$ for some initialisation scale $\alpha > 0$.

- Figure 1 and Figure 4 (Left): $(n, d, \alpha) = (5, 7, 10^{-120})$, $H = I_d$, $\beta^\star = (10, 20, 0, 0, 0, 0, 0) \in \mathbb{R}^7$.

- Figure 4 (Right): $(n, d, \alpha) = (6, 6, 10^{-10})$, $H = \mathrm{diag}(1, 10, 10, 10, 10, 10) \in \mathbb{R}^{6\times6}$, $\beta^\star = (1, 0, 0, 0, 0, 0, 0) \in \mathbb{R}^6$.

- Figure 3 (Left): $(n, d, \alpha_1, \alpha_2) = (7, 2, 10^{-100}, 10^{-10})$, $H = I_d$, $\beta^\star = (10, 20) \in \mathbb{R}^7$.

- Figure 3 (Right): $(n, d, \alpha) = (3, 3, 10^{-100})$ , $X$ is the square root matrix of the matrix $((20, 6, -1.4), (6, 2, -0.4), (-1.4, -0.4, 0.12)) \in \mathbb{R}^{3\times3}$, $\beta^\star = (1, 9, 10)$.

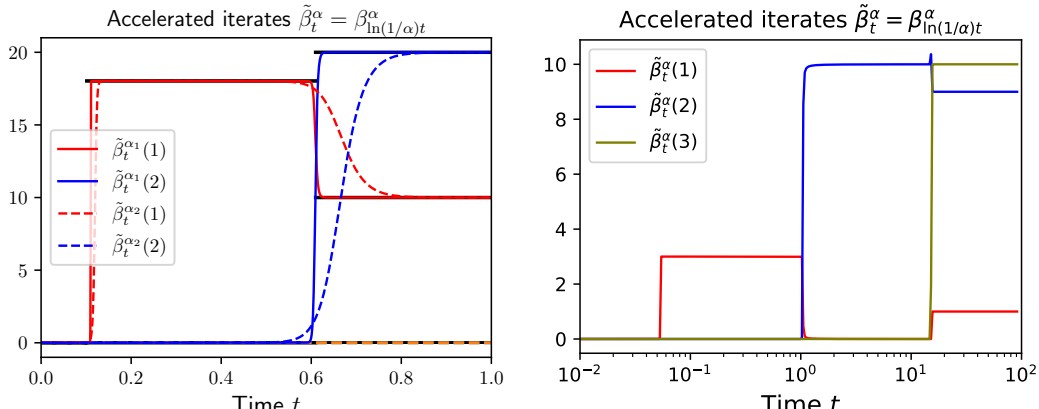

Figure 3: *Left:* Visualisation of the uniform convergence of $\tilde{\beta}^\alpha$ towards $\tilde{\beta}^\circ$ as $\alpha \to 0$. $\alpha_1 = 10^{-100} \ll \alpha_2 = 10^{-10}$ *Right:* In some cases, 2 coordinates can activate at the same time. Note that the time axis is in log-scale for better visualisation.

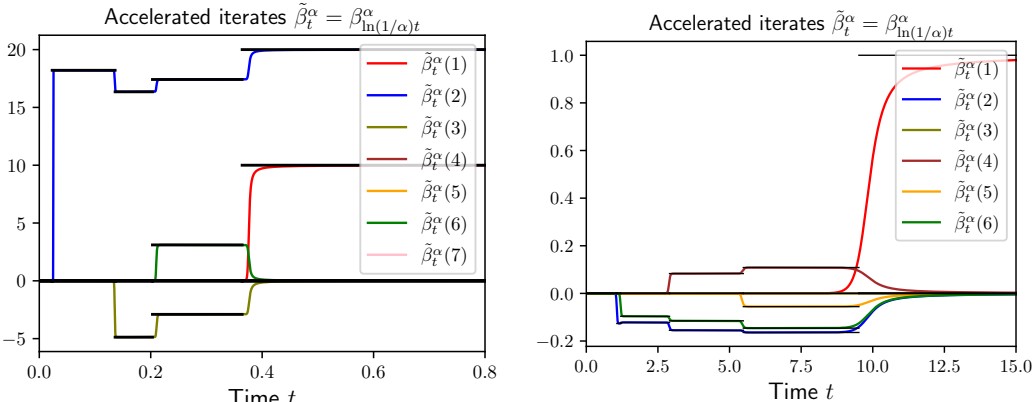

Figure 4: Complex dynamics can occur. *Left and right:* Coordinates are not monotonic and the number of active coordinates neither as several coordinates can deactivate at the same time. The piecewise constant process plotted in black is the limiting process $\tilde{\beta}^\circ$ predicted by our theory.

# B  Proof of Proposition 1

**Proposition 1.** *All the critical points $w_c$ of $F$ which are not global minima, i.e., $\nabla F(w_c) = \mathbf{0}$ and $F(w_c) > \min_w F(w)$, are necessarily saddle points* (i.e., *not local extrema*). *They map to parameters $\beta_c = u_c \odot v_c$ which satisfy $|\beta_c| \odot \nabla L(\beta_c) = \mathbf{0}$ and:*

$$\beta_c \in \underset{\beta[i]=0 \text{ for } i \notin \text{supp}(\beta_c)}{\arg\min} L(\beta) \tag{4}$$

*where $\text{supp}(\beta_c) = \{i \in [d], \beta_c[i] \neq 0\}$ corresponds to the support of $\beta_c$.*

*Proof.* **Non-existence of maxima / non-global minima.** This is a simpler version of results which appear in [30], for the sake of completeness we provide here a simple proof adapted to our setting. The intuition follows the fact that if there existed a local maximum / non-global minimum for $F$ then this would translate to the existence of a local maximum / non-global minimum for the convex loss $L$, which is absurd.

Assume that there exists a local maximum $w^\star = (u^\star, v^\star)$, i.e. assume that there exists $\varepsilon > 0$ such that for all $w = (u, v)$ such that $\|w - w^\star\|_2^2 \leq \varepsilon$, $F(w) \leq F(w^\star)$. We show that this would imply that $\beta^\star = u^\star \odot v^\star$ is a local maximum of $L$, which is absurd.

The mapping $g : (u, v) \mapsto (u \odot v, \sqrt{(u^2 - v^2)/2})$ from $\mathbb{R}_{\geq 0}^d \times \mathbb{R}^d \to \mathbb{R}^d \times \mathbb{R}_{\geq 0}^d$ is a bijection with inverse

$$g^{-1} : (\beta, \alpha) \mapsto (\sqrt{\alpha^2 + \sqrt{\beta^2 + \alpha^4}}, \text{sign}(\beta) \odot \sqrt{-\alpha^2 + \sqrt{\beta^2 + \alpha^4}}). \tag{14}$$

Also notice that $F(g^{-1}(\beta, \alpha)) = L(\beta)$ for all $\beta$ and $\alpha$. Now let $\tilde{\varepsilon} > 0$ and let $\beta \in \mathbb{R}^d$ such that $\|\beta - \beta^\star\|_2^2 \leq \tilde{\varepsilon}$, then for $(u, v) = g^{-1}(\beta, \alpha_*)$ where $\alpha_* = \sqrt{((u^\star)^2 - (v^\star)^2)/2}$ we have that:

$$
\begin{aligned}
\|(u, v) - (u^\star, v^\star)\|_2^2 &= 2\left\|\left(\sqrt{\alpha_*^4 + \beta^2} - \sqrt{\alpha_*^4 + \beta^{\star 2}}\right)^2\right\|_1 \\
&\leq 2\|\beta^2 - \beta^{\star 2}\|_1 \\
&= 2\|(\beta - \beta^\star)^2 + 2(\beta - \beta^\star)\beta^\star\|_1 \\
&\leq 2\|(\beta - \beta^\star)^2\|_1 + 2\|\beta^\star\|_\infty \|\beta - \beta^\star\|_1 \\
&\leq 2(1 + \sqrt{d}\|\beta^\star\|_\infty)\tilde{\varepsilon} \\
&\leq \varepsilon
\end{aligned}
$$

where the last inequality is for $\tilde{\varepsilon}$ small enough. This means that $L(\beta) = F(w) \leq F(w^\star) = L(\beta^\star)$ and $\beta^\star$ is a local maximum of $L$, which is absurd.

The exact same proof holds to show that there are no local minima of $F$ which are not global minima.

**Critical points.** The gradient of the loss function $F$ writes:

$$\nabla_w F(w) = \begin{pmatrix} \nabla_u F(w) \\ \nabla_v F(w) \end{pmatrix} = \begin{pmatrix} \nabla L(\beta) \odot v \\ \nabla L(\beta) \odot u \end{pmatrix} \in \mathbb{R}^{2d}.$$

Therefore $\nabla F(w_c) = \mathbf{0} \in \mathbb{R}^{2d}$ implies that $\nabla L(\beta_c) \odot \beta_c = \mathbf{0} \in \mathbb{R}^d$. Now consider such a $\beta_c$ and let $\text{supp}(\beta_c) = \{i \in [d] \text{ such that } \beta_c(i) \neq 0\}$ denote the support of $\beta_c$. Since $[\nabla L(\beta_c)]_i = 0$ for $i \notin \text{supp}(\beta_c)$, we can therefore write that

$$\beta_c \in \underset{\beta_i=0 \text{ for } i \notin \text{supp}(\beta_c)}{\arg\min} L(\beta).$$

Furthermore we point out that since $\text{supp}(\beta_c) \subset [d]$, there are at most $2^d$ distinct sets $\text{supp}(\beta_c)$, and therefore at most $2^d$ values $F(w_c) = L(\beta_c)$, where $w_c$ is a critical point of $F$. $\qquad \square$

**Additional comment concerning the uniqueness of $\arg\min_{\beta_i=0, i \notin \text{supp}(\beta_c)} L(\beta)$.**

We point out that the constrained minimisation problem (4) does not necessarily have a unique solution, even when $\beta_c$ is not a global solution. Though not required for any of our results, for the sake of completeness, we show here that under an additional mild assumption on the data, we can ensure that the minimisation problem (4) which appears in Proposition 1 has a unique minimum when $L(\beta_c) > 0$. Under this additional assumption, there is therefore a finite number of saddles $\beta_c$. Recall that we let $X \in \mathbb{R}^{n \times d}$ be the feature matrix and $(\tilde{x}_1, \ldots, \tilde{x}_d)$ be its columns. Now assume *temporarily* that the following assumption holds.

**Assumption 2** (Assumption used just in this short section). *Any subset of $(\tilde{x}_1, \ldots, \tilde{x}_d)$ of size smaller than $\min(n, d)$ is linearly independent.*

One can easily check that this assumption holds with probability 1 as soon as the data is drawn from a continuous probability distribution, similarly to [47, Lemma 4]). In the following, for a subset $\xi = \{i_1, \ldots, i_k\} \subset [d]$, we write $X_\xi = (\tilde{x}_{i_1}, \ldots, \tilde{x}_{i_k}) \in \mathbb{R}^{n \times k}$ (we extract the columns from $X$). For a vector $\beta \in \mathbb{R}^d$ we write $\beta[\xi] = (\beta_{i_1}, \ldots, \beta_{i_k})$ and $\beta[\xi^C] = (\beta_i)_{i \notin \xi}$. We distinguish two different settings:

- Underparametrised setting ($n \geq d$) : in this case, for any $\xi = \{i_1, \ldots, i_k\} \subset [d]$, then $\beta^\star := \underset{\beta_i = 0, i \notin \xi}{\mathrm{argmin}} \ L(\beta)$ is unique. Indeed we simply set the gradient to 0 and notice that due to Assumption 2, there exists a unique solution, indeed it is $\beta^\star$ such that $\beta^\star[\xi] = (X_\xi^\top X_\xi)^{-1} X_\xi^\top y$ and $\beta^\star[\xi^C] = 0$.

- Overparametrised setting ($d > n$) : **Global solutions:** $\arg\min_{\beta \in \mathbb{R}^d} L(\beta)$ is an affine space spanned by the orthogonal of $(x_1, \ldots, x_n)$ in $\mathbb{R}^d$. Since $\mathrm{span}(\tilde{x}_1, \ldots, \tilde{x}_d) = \mathbb{R}^n$ from Assumption 2, any $\beta^\star \in \arg\min_{\beta \in \mathbb{R}^d} L(\beta)$ satisfies $X\beta^\star = y$ and $L(\beta^\star) = 0$. **"Saddle points":** now let $\beta_c \in \mathbb{R}^d$ be such that we can write $\beta_c \in \arg\min_{\beta_i = 0, i \notin \mathrm{supp}(\beta_c)} L(\beta)$ and assume that $L(\beta_c) > 0$ (i.e., not a global solution), then: (1) $\beta_c$ has at most $n$ non-zero entries, indeed if it were not the case, then $y$ would necessarily belong to $\mathrm{span}(\tilde{x}_i)_{i \in \mathrm{supp}(\beta_c)}$ due to the assumption on the data, and this would lead to $L(\beta_c) = 0$, (2) therefore, similar to the underparametrised case, $\arg\min_{\beta_i = 0, i \notin \mathrm{supp}(\beta_c)} L(\beta)$ is unique, equal to $\beta_c$, and we have that $\beta_c[\xi] = (X_\xi^\top X_\xi)^{-1} X_\xi^\top y$ and $\beta_c[\xi^C] = 0$ where $\xi = \mathrm{supp}(\beta_c)$.

Thus, in both the underparametrised and overparametrised settings, the minimisation problem (4) appearing in Proposition 1 has a unique minimum when $L(\beta_c) > 0$ and Assumption 2 holds.

## C  General results on the iterates

In the following lemma we recall a few results concerning the gradient flow Eq. (5):

$$\mathrm{d}w_t = -\nabla F(w_t)\mathrm{d}t\,, \tag{15}$$

where $F$ is defined in Eq. (3) as:

$$F(w) \coloneqq L(u \odot v) = \frac{1}{2n}\sum_{i=1}^{n}(\langle u \odot v, x_i\rangle - y_i)^2\,.$$

**Lemma 1.** *For an initialisation $u_0 = \sqrt{2}\alpha$, $v_0 = \mathbf{0}$, the flow $w_t^\alpha = (u_t^\alpha, v_t^\alpha)$ from Eq. (15) is such that the quantity $(u_t^\alpha)^2 - (v_t^\alpha)^2$ is constant and equal to $2\alpha^2\mathbf{1}$. Furthermore $u_t^\alpha > |v_t^\alpha| \geq 0$ and therefore from the bijection Eq. (14) we have that:*

$$u_t^\alpha = \sqrt{\alpha^2 + \sqrt{(\beta_t^\alpha)^2 + \alpha^4}}, \quad v_t^\alpha = \mathrm{sign}(\beta_t^\alpha) \odot \sqrt{-\alpha^2 + \sqrt{(\beta_t^\alpha)^2 + \alpha^4}}.$$

*Proof.* From the expression of $\nabla F(w)$, notice that the derivative of $(u_t^\alpha)^2 - (v_t^\alpha)^2$ is equal to $\mathbf{0}$ and therefore equal to its initial value.

Since $(u_t^\alpha)^2 - (v_t^\alpha)^2 = (u_t^\alpha + v_t^\alpha)(u_t^\alpha - v_t^\alpha) > 0$, by continuity we get that $u_t^\alpha + v_t^\alpha > 0$ and $u_t^\alpha - v_t^\alpha > 0$ and therefore $u_t^\alpha > |v_t^\alpha|$. □

In this section we consider the accelerated iterates Eq. (9) which follow:

$$\mathrm{d}\nabla\tilde{\phi}_\alpha(\tilde{\beta}_t^\alpha) = -\nabla L(\tilde{\beta}_t^\alpha)\mathrm{d}t, \qquad \text{where} \qquad \tilde{\phi}_\alpha \coloneqq \frac{1}{\ln(1/\alpha)} \cdot \tilde{\phi}_\alpha \tag{16}$$

with $\tilde{\beta}_{t=0} = \mathbf{0}$ and where $\phi_\alpha$ is defined Eq. (7).

**Proposition 4.** *For all $\alpha > 0$ and minimum $\beta^\star \in \arg\min_\beta L(\beta)$, the loss values $L(\tilde{\beta}_t^\alpha)$ and the Bregman divergence $\mathrm{D}_{\tilde{\phi}_\alpha}(\beta^\star, \tilde{\beta}_t^\alpha)$ are decreasing. Moreover*

$$L(\tilde{\beta}_t^\alpha) - L(\beta^\star) \leq \frac{\tilde{\phi}_\alpha(\beta^\star)}{2t}, \tag{17}$$

$$L\Big(\frac{1}{t}\int_0^t \tilde{\beta}_s^\alpha \mathrm{d}s\Big) - L(\beta^\star) \leq \frac{\tilde{\phi}_\alpha(\beta^\star)}{2t}. \tag{18}$$

*Proof.* The loss is decreasing since: $\frac{\mathrm{d}}{\mathrm{d}t}L(\tilde{\beta}_t^\alpha) = \nabla L(\tilde{\beta}_t^\alpha)^\top \dot{\tilde{\beta}}_t^\alpha = -\dot{\tilde{\beta}}_t^{\alpha\top}\nabla^2\tilde{\phi}_\alpha(\tilde{\beta}_t^\alpha)\dot{\tilde{\beta}}_t^\alpha \leq 0$.

$\frac{\mathrm{d}}{\mathrm{d}t}\mathrm{D}_{\tilde{\phi}_\alpha}(\beta^\star, \tilde{\beta}_t^\alpha) = -\nabla L(\tilde{\beta}_t^\alpha)^\top(\tilde{\beta}_t^\alpha - \beta^\star) = -2(L(\tilde{\beta}_t^\alpha) - L(\beta^\star))$ (since $L$ is the quadratic loss), therefore the Bregman distance is decreasing. We can also integrate this last equality from $0$ to $t$, and divide by $-2t$:

$$\frac{1}{t}\int_0^t L(\tilde{\beta}_s^\alpha)\mathrm{d}s - L(\beta^\star) = \frac{\mathrm{D}_{\tilde{\phi}_\alpha}(\beta^\star, \beta_0^\alpha = \mathbf{0}) - \mathrm{D}_{\tilde{\phi}_\alpha}(\beta^\star, \beta_t^\alpha)}{2t}$$

$$\leq \frac{\tilde{\phi}_\alpha(\beta^\star)}{2t}.$$

Since the loss is decreasing we get that $L(\tilde{\beta}_t^\alpha) - L(\beta^\star) \leq \frac{\tilde{\phi}_\alpha(\beta^\star)}{2t}$ and from the convexity of $L$ we get that $L\Big(\frac{1}{t}\int_0^t \tilde{\beta}_s^\alpha \mathrm{d}s\Big) - L(\beta^\star) \leq \frac{\tilde{\phi}_\alpha(\beta^\star)}{2t}$. □

In the following proposition, we show that for $\alpha$ small enough, the iterates are bounded independently of $\alpha$. Note that this result unfortunately only holds for the quadratic loss, we expect it to hold for other convex losses of the type $L(\beta) = \frac{1}{n}\sum_i \ell(y_i, \langle x_i, \beta\rangle)$ where $\ell(y, \cdot)$ is strictly convex has a unique root at $y$ but we don't know how to show it. Also note that bounding the accelerated iterates $\tilde{\beta}^\alpha$ is equivalent to bounding the iterates $\beta^\alpha$ since $\tilde{\beta}_t^\alpha = \beta_{\ln(1/\alpha)t}^\alpha$.

**Proposition 5.** *For $\alpha < \alpha_0$, where $\alpha_0$ depends on $\beta_{\ell_1}^\star$, the iterates $\tilde{\beta}_t^\alpha$ are bounded independently of $\alpha$:*

$$\|\tilde{\beta}_t^\alpha\|_\infty \leq 3\|\beta_{\ell_1}^\star\|_1 + 1$$

*Proof.* From Eq. (16), integrating and using that $L$ is the quadratic loss, we get:

$$\nabla\tilde{\phi}_\alpha(\tilde{\beta}_t^\alpha) = \frac{t}{n}X^\top(y - X\bar{\beta}_t^\alpha) = -\frac{t}{n}X^\top X(\bar{\beta}_t^\alpha - \beta^\star),$$

where we recall that $X \in \mathbb{R}^{n \times d}$ is the input data represented as a matrix and where we denote the averaged iterate by $\bar{\beta}_t^\alpha = \frac{1}{t}\int_0^t \tilde{\beta}_s^\alpha \mathrm{d}s$. Thus we get

$$\nabla\tilde{\phi}_\alpha(\tilde{\beta}_t^\alpha)^\top(\tilde{\beta}_t^\alpha - \beta^\star) = -\frac{t}{n}(\bar{\beta}_t^\alpha - \beta^\star)^\top X^\top X(\tilde{\beta}_t^\alpha - \beta^\star). \tag{19}$$

By convexity of $\tilde{\phi}_\alpha$ we have $\tilde{\phi}_\alpha(\beta_t^\alpha) - \tilde{\phi}_\alpha(\beta^\star) \leq \nabla\tilde{\phi}_\alpha(\beta_t^\alpha)^\top(\beta_t^\alpha - \beta^\star)$. By the Cauchy-Schwarz inequality, we also have $(\bar{\beta}_t^\alpha - \beta^\star)^\top X^\top X(\beta_t^\alpha - \beta^\star) \leq \|X(\beta_t^\alpha - \beta^\star)\|\|X(\bar{\beta}_t^\alpha - \beta^\star)\|$. Using Proposition 4: $\|X(\beta_t^\alpha - \beta^\star)\|^2 \leq n\tilde{\phi}_\alpha(\beta^\star)/t$ and $\|X(\bar{\beta}_t^\alpha - \beta^\star)\|^2 \leq n\tilde{\phi}_\alpha(\beta^\star)/t$ we can further bound the right hand side of Eq. (19) as

$$-\frac{t}{n}(\bar{\beta}_t^\alpha - \beta^\star)^\top X^\top X(\beta_t^\alpha - \beta^\star) \leq \tilde{\phi}_\alpha(\beta^\star).$$

Thus it yields

$$\tilde{\phi}_\alpha(\beta_t^\alpha) - \tilde{\phi}_\alpha(\beta^\star) \leq \tilde{\phi}_\alpha(\beta^\star).$$

From [50] (proof of Lemma 1 in the appendix) we get that for

$$\alpha < \min\left\{1, \sqrt{\|\beta\|_1}, (2\|\beta\|_1)^{-1}\right\}$$

then:

$$\tilde{\phi}_\alpha(\beta) \leq \frac{3}{2}\|\beta\|_1,$$

and for all $\alpha < \exp(-d/2)$:

$$\tilde{\phi}_\alpha(\beta) \geq \|\beta\|_1 - \frac{d}{\ln(1/\alpha^2)}$$
$$\geq \|\beta\|_1 - 1,$$

which finally leads for

$$\alpha < \alpha_0 := \min\left\{1, \sqrt{\|\beta_{\ell_1}^\star\|_1}, (2\|\beta_{\ell_1}^\star\|_1)^{-1}, \exp(-d/2)\right\}$$

to the result. $\qquad\square$

The following proposition shows that we can bound the path length of the flow $\tilde{\beta}^\alpha$ independently of $\alpha$. Keep in mind that the path length of $\tilde{\beta}^\alpha$ is equivalent to that of $\beta^\alpha$ as the first is just an acceleration of the second: $\tilde{\beta}_t^\alpha = \beta_{\ln(1/\alpha)t}^\alpha$.

**Proposition 6.** *For $\alpha < \alpha_0$ where $\alpha_0$ is the same as in Proposition 5, the path length of the iterates $(\beta_t^\alpha)_{t \geq 0}$ is bounded independently of $\alpha > 0$:*

$$\int_0^{+\infty} \|\dot{\beta}_t^\alpha\|\mathrm{d}t < C,$$

*where $C$ does not depend on $\alpha$. Hence the path length of the accelerated flow $\tilde{\beta}^\alpha$ is also bounded independently of $\alpha$.*

*Proof.* Having shown that the iterates $\beta_t^\alpha$ are bounded independently of $\alpha$, it also implies that the iterates $w_t = (u_t, v_t)$ are bounded following Lemma 1. Since the loss $w \mapsto F(w)$ is a multivariate polynomial function, it is a semialgebraic function and we can consequently apply the result of Kurdyka [32, Theorem 2] which grants that

$$\int_0^{+\infty} \|\dot{w}_t\| \mathrm{d}t < C,$$

where the constant $C$ only depends on the loss and on the bound on the iterates. We further use that $\dot{\beta} = \dot{u} \odot v + u \odot \dot{v}$ and $\|\dot{u} \odot v + u \odot \dot{v}\| \le C_1 (\|\dot{u}\| + \|\dot{v}\|)$ using that $u$ and $v$ are bounded and $\|\dot{u}\| + \|\dot{v}\| \le C_2 \|\dot{w}\|$ using the equivalence of norms. Therefore $\int_0^{+\infty} \|\dot{\beta}_t^\alpha\| \mathrm{d}t < C$ for some $C$ which is independent of the initialisation scale $\alpha$. $\qquad\square$

# D Standalone properties of Algorithm 1

## D.1 "Well-definedness" of Algorithm 1 and upperbound on its number of loops

Notice that this proposition highlights the fact that Algorithm 1 is on its own an algorithm of interest for finding the minimum $\ell_1$-norm solution in an overparametrised regression setting. We point out that the provided upperbound on the number of iterations is very crude and could certainly be improved.

**Proposition 7.** *Algorithm 1 is well defined: at each iteration (i) the attribution of $\Delta$ is well defined as $\Delta < +\infty$, (ii) the constrained minimisation problem has a unique solution and the attribution of the value of $\beta$ is therefore well-founded. Furthermore, along the loops: the iterates $\beta$ have at most $n$ non-zero coordinates, the loss is strictly decreasing and the algorithm terminates in at most $\min\left(2^d, \sum_{k=0}^n \binom{d}{k}\right)$ steps by outputting the minimum $\ell_1$-norm solution $\beta_{\ell_1}^\star := \underset{\beta \in \arg\min L}{\arg\min} \ \|\beta\|_1$.*

*Proof.* In the following, for the matrix $X$ and for a subset $I = \{i_1, \dots, i_k\} \subset [d]$, we write $X_I = (\tilde{x}_{i_1}, \dots, \tilde{x}_{i_k}) \in \mathbb{R}^{n \times k}$ (we extract the columns from $X$). For a vector $\beta \in \mathbb{R}^d$ we write $\beta_I = (\beta_{i_1}, \dots, \beta_{i_k})$.

**(1) The constrained minimisation problem has a unique solution:** we follow the proof of [47, Lemma 2]. Following the notations in Algorithm 1, we define $I = \{i \in [d], |s_i| = 1\}$ and we point out that after $k$ loops of the algorithm, the value of $s$ is equal to $s = -(\Delta_1 \nabla L(\beta_0) + \dots + \Delta_k \nabla L(\beta_{k-1})) \in \mathrm{span}(x_1, \dots, x_n)$. We can therefore write $s = X^\top r$ for some $r \in \mathbb{R}^n$.

Now assume that $\ker(X_I) \neq \{0\}$. Then, for some $i \in I$, we have $\tilde{x}_i = \sum_{j \in I \setminus \{i\}} c_j \tilde{x}_j$ where $c_j \in \mathbb{R}$. Without loss of generality, we can assume that $I \setminus \{i\}$ has at most $n$ elements. Indeed, we can otherwise always find $n$ elements $\tilde{I} \subset I \setminus \{i\}$ such that $\tilde{x}_i = \sum_{j \in \tilde{I}} c_j \tilde{x}_j$. Rewriting the previous equality, we get

$$s_i \tilde{x}_i = \sum_{j \in I \setminus \{i\}} (s_i s_j c_j)(s_j \tilde{x}_j). \tag{20}$$

Now by definitions of the set $I$ and of $r$, we have that $\langle \tilde{x}_j, r \rangle = s_j \in \{+1, -1\}$ for any $j \in I$. Taking the inner product of Eq. (20) with $r$, we obtain that $1 = \sum_{j \in I \setminus \{i\}} (s_i s_j c_j)$. Consequently, we have shown that if $\ker(X_I) \neq \{0\}$, then we necessarily have for some $i \in I$,

$$s_i \tilde{x}_i = \sum_{j \in I \setminus \{i\}} a_j (s_j \tilde{x}_j),$$

with $\sum_{j \in I \setminus \{i\}} a_j = 1$, which means that $s_i \tilde{x}_i$ lies in the affine space generated by $(s_j \tilde{x}_j)_{j \in I \setminus \{i\}}$. This fact is however impossible due to Assumption 1 (recall that without loss of generality we have that $I \setminus \{i\}$ has at most $n$ elements, and trivially less that $d$ elements). **Therefore $X_I$ is full rank**, and $\mathrm{Card}(I) \leq n$. Now notice that the constrained minimisation problem corresponds to $\arg\min_{\substack{\beta_i \geq 0, i \in I_+ \\ \beta_i \leq 0, i \in I_-}} \|y - X_I \beta_I\|_2^2$. Since $X_I$ is full rank, this restricted loss is strictly convex and the constrained minimisation problem **has a unique minimum.**

**(2) $\Delta < +\infty$:** Notice that the optimality conditions of

$$\beta = \underset{\substack{\beta_i \geq 0, i \in I_+ \\ \beta_i \leq 0, i \in I_- \\ \beta_i = 0, i \notin I}}{\arg\min} \|y - X_I \beta_I\|_2^2,$$

are (i) $\beta$ satisfies the constraints, (ii) if $i \in I_+$ (resp $i \in I_-$) then $[-\nabla L(\beta)]_i \leq 0$ (resp $[-\nabla L(\beta)]_i \geq 0$) and (iii) if $\beta_i \neq 0$ then $[\nabla L(\beta)]_i = 0$. One can notice that condition (ii) ensures that at each iteration, for $\delta \leq \Delta_k$, $s_{k-1} - \delta \nabla L(\beta_{k-1}) \in [-1, 1]$ coordinate wise. Also, if $L(\beta_{k-1}) \neq \mathbf{0}$, then a coordinate of the vector $|s_{k-1} - \delta \nabla L(\beta_{k-1})|$ must necessarily hit 1, this value of $\delta$ corresponds to $\Delta_k$.

**(3) The loss is strictly decreasing:** Let $I_{k-1,\pm}$ and $I_{k,\pm}$ be the equicorrelation sets defined in the algorithm at step $k-1$ and $k$, and $\beta_{k-1}$ and $\beta_k$ the solutions of the constrained minimisation problems. Also, let $i_k$ be the newly added coordinate which breaks the constraint at step $k$ (which we assume to be unique for simplicity). Without loss of generality, assume that $s_k(i_k) = +1$. Since the sets

$I_{k-1,+} \setminus (I_{k,+} \setminus \{i_k\})$ and $I_{k-1,-} \setminus I_{k,-}$ are (if not empty) only composed of indexes of coordinates of $\beta_{k-1}$ which are equal to 0, one can notice that $\beta_{k-1}$ also satisfies the new constraints at step $k$. Therefore $L(\beta_k) \leq L(\beta_{k-1})$. Now since $[-\nabla L(\beta_{k-1})]_{i_k} > 0$, from the strict convexity of the restricted loss on $I_k$, this means that $\beta_k(i_k) > 0$ (which also means that newly activated coordinate $i_k$ **must activate**), and therefore $\beta_{k-1} \neq \beta_k$ and $L(\beta_k) < L(\beta_{k-1})$.

**(4) The algorithm terminates in at most** $\min\left(2^d, \sum_{k=0}^n \binom{d}{k}\right)$ **steps:** Recall that we showed in part (1) of the proof that at each iteration $k$ of the algorithm, $I_k$ as at most $\min(n, d)$ elements. Since $\operatorname{supp}(\beta_k) \subset I_k$, we have that $\beta_k$ has at most $\min(n, d)$ non-zero elements, also recall that we always have $\beta_k = \arg\min_{\beta_i = 0, i \notin \operatorname{supp}(\beta_k)} L(\beta)$ (we here have unicity of this minimisation problem following part (1) of the proof). There are hence at most

$$\sum_{k=0}^{\min(n,d)} \binom{d}{k} = \min\left(2^d, \sum_{k=0}^n \binom{d}{k}\right)$$

such minimisation problems. The loss being strictly decreasing, the algorithm cannot output the same solution $\beta$ at two different loops, and the algorithm must terminate in at most $\min\left(2^d, \sum_{k=0}^n \binom{d}{k}\right)$ iterations by outputting a vector $\beta^\star$ such that $\nabla L(\beta^\star) = 0$, *i.e.* $\beta^\star \in \arg\min L(\beta)$.

**(5) The algorithm outputs the minimum $\ell_1$-norm solution.** Let $\beta^\star$ be the output of the algorithm after $p$ iterations. Notice that by the definition of the successive sets $I_{k,\pm}$ and of the constraints on the minimisation problem, we have that at each iteration $s_k \in \partial \|\beta_k\|_1$. Therefore $s_p \in \partial \|\beta^\star\|_1$. Also, recall from part (1) of the proof that $s_p \in \operatorname{span}(x_1, \ldots, x_n)$ which means that there exists $r \in \mathbb{R}^n$ such that $s_p = X^\top r$. Putting the two together we get that $X^\top r \in \partial \|\beta^\star\|_1$, this condition along with the fact that $L(\beta^\star) = \min L(\beta)$ are exactly the KKT conditions of $\arg\min_{\beta \in \arg\min L} \|\beta\|_1$. $\qquad \square$

To put our upperbound on the number of iterations into perspective, the worst-case number of iterations for the LARS algorithm is $(3^d + 1)/2$ [34]. Hence Algorithm 1 has fewer iterations in the worst-case setting. Whether an exponential dependency in the dimension is inevitable for Algorithm 1 is unknown and we leave this as future work.

However, when the number of samples is much smaller than the dimension we lose the exponential dependency. Indeed, for $\varepsilon := n/d \leq 1/2$, we have the upperbound $\sum_{k=0}^n \binom{d}{k} \leq 2^{H(\varepsilon)d}$ where $H(\varepsilon) = -\varepsilon \log_2(\varepsilon) - (1 - \varepsilon) \log_2(1 - \varepsilon)$ is the binary entropy. Since for $\varepsilon \leq 1/2$, $H(\varepsilon) \leq -2\varepsilon \log_2(\varepsilon)$, we get the upperbound $\sum_{k=0}^n \binom{d}{k} \leq 2^{H(\varepsilon)d} \leq (\frac{d}{n})^{2n}$, which is much better than $2^d$.

### D.2 Proof of Proposition 2

As mentioned several times, for general feature matrices $X$ complex behaviours can occur with coordinates deactivating and changing sign several times. Here we show that for simple datasets which have a feature matrix $X$ that satisfy the restricted isometry property (RIP) [14], we can simply determine the jump times and the saddles as a function of the sparse predictor which we seek to recover.

The non-realistic but enlightening extreme case of the RIP assumption is to consider that the feature matrix is such that $X^\top X/n = I_d$. In this case, by letting $\beta^\star$ be the unique vector such that $y = \langle x, \beta^\star \rangle$ and assuming that $\beta^\star = (\beta_1^\star, \ldots, \beta_r^\star, 0, \ldots, 0)$ with $|\beta_1^\star| > \cdots > |\beta_r^\star| > 0$, then the loss writes $L(\beta) = \|\beta - \beta^\star\|_2^2/2$ and one can easily check that Algorithm 1 would terminate in $r$ loops and output exactly $t_i = \frac{1}{|\beta_i^\star|}$ and $\beta_i = (\beta_1^\star, \ldots, \beta_i^\star, 0, \ldots, 0)$ for $i \leq r$ (the case where several coordinates of $\beta^\star$ are stricly equal can also be treated: for example if $\beta_1^\star = \beta_2^\star$ then the first output of the algorithm is directly $\beta_1 = (\beta_1^\star, \beta_2^\star, 0, \ldots, 0)$).

We now recall the more realistic RIP setting which is an adaptation of the previous observation.

**Sparse regression with RIP and gap assumption.** *(RIP) Assume that there exists an $r$-sparse vector $\beta^\star$ such that $y_i = \langle x_i, \beta^\star \rangle$. Furthermore we assume that the feature matrix $X \in \mathbb{R}^{n,d}$ satisfies the $2r$-restricted isometry property with constant $\tilde{\varepsilon} < \sqrt{2} - 1 < 1/2$: i.e. for all submatrix $X_s$ where we extract any $s \leq 2r$ columns of $X$, the matrix $X_s^\top X_s/n$ of size $s \times s$ has all its eigenvalues in the interval $[1 - \tilde{\varepsilon}, 1 + \tilde{\varepsilon}]$. (Gap assumption) Furthermore we assume that the $r$-sparse vector $\beta^\star$*

has coordinates which have a "sufficient gap'. W.l.o.g we write $\beta^\star = (\beta_1^\star, \ldots, \beta_r^\star, 0, \ldots, 0)$ with $|\beta_1^\star| \geq \ldots \geq |\beta_r^\star| > 0$ and we define $\lambda := \min_{i \in [r]}(|\beta_i^\star| - |\beta_{i+1}^\star|) \geq 0$ which corresponds to the smallest gap between the entries of $|\beta^\star|$. We assume that $5\tilde{\varepsilon}\|\beta^\star\|_2 < \lambda/2$ and we let $\varepsilon := 5\tilde{\varepsilon}$.

A classic result from compressed sensing (see Candes [13, Theorem 1.2]) is that the $2r$-restricted isometry property with constant $\sqrt{2} - 1$ ensures that the minimum $\ell_0$-minimisation problem has a unique $r$-sparse solution which is $\beta^\star$. Furthermore it ensures that the minimum $\ell_1$-norm solution is unique and is equal to $\beta^\star$. This means that Algorithm 1 will have $\beta^\star$ as a final output.

We now recall the result which characterises the outputs of Algorithm 1 when the data satisfies the previous assumptions.

**Proposition 2.** *Under the restricted isometry property and the gap assumption stated right above, Algorithm 1 terminates in $r$-loops and outputs:*

$$\beta_1 = (\beta_1[1], 0, \ldots, 0) \qquad \qquad \text{with} \qquad \beta_1[1] \in [\beta_1^\star - \varepsilon\|\beta^\star\|, \beta_2^\star + \varepsilon\|\beta^\star\|]$$

$$\beta_2 = (\beta_2[1], \beta_2[2], 0, \ldots, 0) \qquad \text{with} \quad \begin{cases} \beta_2[1] \in [\beta_1^\star - \varepsilon\|\beta^\star\|, \beta_1^\star + \varepsilon\|\beta^\star\|] \\ \beta_2[2] \in [\beta_2^\star - \varepsilon\|\beta^\star\|, \beta_2^\star + \varepsilon\|\beta^\star\|] \end{cases}$$

$$\vdots$$

$$\beta_{r-1} = (\beta_{r-1}[1], \ldots, \beta_{r-1}[r-1], 0, \ldots, 0) \quad \text{with} \ \beta_{r-1}[i] \in [\beta_i^\star - \varepsilon\|\beta^\star\|, \beta_i^\star + \varepsilon\|\beta^\star\|]$$

$$\beta_r = \beta^\star = (\beta_1^\star, \ldots, \beta_r^\star, 0, \ldots, 0),$$

*at times $t_1, \ldots, t_r$ such that $t_i \in \left[\frac{1}{|\beta_i^\star| + \varepsilon\|\beta^\star\|}, \frac{1}{|\beta_i^\star| - \varepsilon\|\beta^\star\|}\right]$ and where $\|\cdot\|$ denotes the $\ell_2$ norm.*

*Proof.* In all the proof $\|\cdot\|$ denotes the $\ell_2$ norm $\|\cdot\|_2$. For simplicity we assume that $\beta_i^\star > 0$ for all $i \in [r]$, the proof can easily be adapted to the general case. We first define $\xi := X^\top X/n - I_d$. By the restricted isometry property, for any $k \leq 2r$, we have that any $k \times k$ square matrix extracted from $\xi$ which we denote $\xi_{kk}$ has its eigenvalues in $[-\tilde{\varepsilon}, \tilde{\varepsilon}]$. It also means that the eigenvalues of $(I_k + \xi_{kk})^{-1} - I_k$ are in $[\frac{1}{1+\tilde{\varepsilon}} - 1, \frac{1}{1-\tilde{\varepsilon}} - 1] \subset [-2\tilde{\varepsilon}, 2\tilde{\varepsilon}]$.

We now proceed by induction with the following induction hypothesis:

- $\beta_{k-1}$ has its support on its $(k-1)$ first coordinates with $|\beta_{k-1}[i] - \beta_i^\star| \leq 5\tilde{\varepsilon}\|\beta^\star\|$ for $i < k$

- $t_k \in \left[\frac{1}{\beta_k^\star + 5\tilde{\varepsilon}\|\beta^\star\|}, \frac{1}{\beta_k^\star - 5\tilde{\varepsilon}\|\beta^\star\|}\right]$ and $s_{t_k}[k] = 1$

- $s_{t_k}[i] \in [t_k(\beta_i^\star - 5\tilde{\varepsilon}\|\beta^\star\|), t_k(\beta_i^\star + 5\tilde{\varepsilon}\|\beta^\star\|)] \subset (-1, 1)$ for $i > k$

From the recurrence hypothesis, the output of the algorithm at step $k$ is hence $\beta_k = \arg\min L(\beta)$ under the constraint $\beta[i] \geq 0$ for $i \leq k$ and $\beta[i] = 0$ otherwise. We first search for the solution of the minimisation problem without the sign constraint and still (abusively) denote it $\beta_k$: we will show that it turns out to satisfy the sign constraint and that it is therefore indeed $\beta_k$.

In the following, for a vector $v$, we denote by $v[:k]$ its $k$ first coordinates. Setting the $k$ first coordinates of the gradient to 0, we get that $[X^\top X(\beta_k - \beta^\star)][:k] = \mathbf{0}$, which leads to $(I_k + \xi_{kk})\beta_k[:k] = \beta^\star[:k] + [\xi\beta^\star][:k]$, which gives:

$$\beta_k[:k] = (I_k + \xi_{kk})^{-1}(\beta^\star[:k] + [\xi\beta^\star][:k])$$
$$= \beta^\star[:k] + [\xi\beta^\star][:k] + v_1$$

where from the bound on the eigenvalues of $(I_k + \xi_{kk})^{-1} - I_k$ and $\|\xi\beta^\star\| \leq \tilde{\varepsilon}\|\beta^\star\|$:

$$\|v_1\| \leq 2\tilde{\varepsilon}\|\beta^\star[:k] + [\xi\beta^\star][:k])\|$$
$$\leq 2\tilde{\varepsilon}(\|\beta^\star\| + \|\xi\beta^\star\|)$$
$$\leq 2\tilde{\varepsilon}(\|\beta^\star\| + \tilde{\varepsilon}\|\beta^\star\|)$$
$$\leq 4\tilde{\varepsilon}\|\beta^\star\|.$$

Therefore

$$\beta_k[:k] = \beta^\star[:k] + v_2$$

where $v_2 = [\xi\beta^\star][: k] + v_1$ hence $\|v_2\|_\infty \leq \|v_2\| \leq 5\tilde{\varepsilon}\|\beta^\star\|$. Notice that from the definition of $\lambda$ and the fact that $5\tilde{\varepsilon}\|\beta^\star\| < \lambda/2$ we have that $\beta_k[:k] \geq 0$ coordinate-wise, hence verifying the sign constraint. Also note that $\|\beta_k\| \leq \|\beta^\star\| + 5\tilde{\varepsilon}\|\beta^\star\| \leq 4\|\beta^\star\|$.

For $t \geq t_k$, $s_t = s_{t_k} - (t - t_k)\nabla L(\beta_k)$, and $[\nabla L(\beta_k)][: k] = 0$ therefore $s_t[: k] = s_{t_k}[: k]$. Now for $i > k$, $[-\nabla L(\beta_k)]_i = n^{-1}[X^\top X(\beta^\star - \beta_k)]_i = \beta_i^\star + [\xi(\beta_k - \beta^\star)]_i$. Now since $(\beta_k - \beta^\star)$ is $r$-sparse we have that:

$$
\begin{aligned}
\|\xi(\beta_k - \beta^\star)\|_\infty &\leq \|\xi(\beta_k - \beta^\star)\| \\
&\leq \tilde{\varepsilon}\|\beta_k - \beta^\star\| \\
&\leq \tilde{\varepsilon}(\|\beta_k\| + \|\beta^\star\|) \\
&\leq 5\tilde{\varepsilon}\|\beta^\star\| < \lambda/2,
\end{aligned}
\tag{21}
$$

Now from the fact that $s_t[i] = s_{t_k}[i] + (t - t_k)\beta_i^\star + (t - t_k)[\xi(\beta_k - \beta^\star)]_i$ and using the recurrence hypothesis: $s_{t_k}[i] \in [t_k(\beta_i^\star - 5\tilde{\varepsilon}\|\beta^\star\|), t_k(\beta_i^\star + 5\tilde{\varepsilon}\|\beta^\star\|)]$, we get (using the bound Eq. (21)) that $s_t[i] \in [t(\beta_i^\star - 5\tilde{\varepsilon}\|\beta^\star\|), t(\beta_i^\star + 5\tilde{\varepsilon}\|\beta^\star\|)]$. From the "separation assumption" we have that $5\tilde{\varepsilon}\|\beta^\star\| < \lambda/2$ and therefore the next coordinate to activate is necessarily the $(k+1)^{th}$ at time $t_{k+1}$ with $s_{t_{k+1}}[k+1] = 1$ and:

$$
t_{k+1} \in \left[ \frac{1}{\beta_{k+1}^\star + 5\tilde{\varepsilon}\|\beta^\star\|}, \frac{1}{\beta_{k+1}^\star - 5\tilde{\varepsilon}\|\beta^\star\|} \right].
$$

This proves the recursion. The algorithm cannot stop before iteration $r$ as $\beta^\star$ is the unique minimiser of $L$ that has at most $r$ non-zero coordinates. But it stops at iteration $r$ as $\beta^\star$ is the unique minimiser of $L(\beta)$ under the constraints $\beta_i \geq 0$ for $i \leq r$ and $\beta_i = 0$ otherwise. $\qquad\square$

# E   Proof of Theorem 2 and Proposition 3 through the arc-length parametrisation

In this section, we explain in more details the arc-length reparametrisation which circumvents the apparition of discontinuous jumps and leads to the proof of Theorem 2. The main difficulty to show the convergence stems from the non-continuity of the limit process $\tilde{\beta}^\circ$. Therefore we cannot expect uniform convergence of $\tilde{\beta}^\alpha$ towards $\tilde{\beta}$ as $\alpha \to 0$. In addition, $\tilde{\beta}^\circ$ does not provide any insights into the path followed between the jumps.

**Arc-length parametrisation.** The high-level idea is to "slow-down" time when the jumps occur. To do so we follow the approach from [18, 36] and we consider an arc-length parametrisation of the path, i.e., we consider $\tau^\alpha$ equal to:

$$\tau^\alpha(t) = t + \int_0^t \|\dot{\tilde{\beta}}_s^\alpha\| \mathrm{d}s.$$

In Proposition 6, we showed that the full path length $\int_0^{+\infty} \|\dot{\beta}_s^\alpha\| \mathrm{d}s$ is finite and bounded independently of $\alpha$. Therefore $\tau^\alpha$ is a bijection in $\mathbb{R}_{\geq 0}$. We can then define the following quantities:

$$\hat{t}_\tau^\alpha = (\tau^\alpha)^{-1}(\tau) \quad \text{and} \quad \hat{\beta}_\tau^\alpha = \tilde{\beta}_{\hat{t}^\alpha(\tau)}^\alpha.$$

By construction, a simple chain rule leads to $\dot{\hat{t}}^\alpha(\tau) + \|\dot{\hat{\beta}}_\tau^\alpha\| = 1$, which means that the speed of $(\hat{\beta}_\tau^\alpha)_\tau$ is always upperbounded by 1, independently of $\alpha$. This behaviour is in stark contrast with the process $(\tilde{\beta}_t^\alpha)_t$ which has a speed which explodes at the jumps. It presents a major advantage as we can now use Arzelà-Ascoli's theorem to extract a converging subsequent. A simple change of variable shows that the new process satisfies the following equations:

$$-\int_0^\tau \dot{\hat{t}}_s^\alpha \nabla L(\hat{\beta}_s^\alpha) \mathrm{d}s = \nabla \tilde{\phi}_\alpha(\hat{\beta}_\tau^\alpha) \quad \text{and} \quad \dot{\hat{t}}_\tau^\alpha + \|\dot{\hat{\beta}}_\tau^\alpha\| = 1 \tag{22}$$

started from $\hat{\beta}_\tau^\alpha = 0$ and $\hat{t}_0 = 0$. The next proposition states the convergence of the rescaled process, up to a subsequence.

**Proposition 8.** *Let $T \geq 0$. For every $\alpha > 0$, let $(\hat{t}^\alpha, \hat{\beta}^\alpha)$ be the solution of Eq. (22). Then, there exists a subsequence $(\hat{t}^{\alpha_k}, \hat{\beta}^{\alpha_k})_{k \in \mathbb{N}}$ and $(\hat{t}, \hat{\beta})$ such that as $\alpha_k \to 0$ :*

$$(\hat{t}^{\alpha_k}, \hat{\beta}^{\alpha_k}) \to (\hat{t}, \hat{\beta}) \qquad \text{in } (C^0([0,T], \mathbb{R} \times \mathbb{R}^d), \|\cdot\|_\infty) \tag{23}$$

$$(\dot{\hat{t}}^{\alpha_k}, \dot{\hat{\beta}}^{\alpha_k}) \rightharpoonup (\dot{\hat{t}}, \dot{\hat{\beta}}) \qquad \text{in } L_1[0,T] \tag{24}$$

**Limiting dynamics.** *The limits $(\hat{t}, \hat{\beta})$ satisfy:*

$$-\int_0^\tau \dot{\hat{t}}_s \nabla L(\hat{\beta}_s) \mathrm{d}s \in \partial \|\hat{\beta}_\tau\|_1 \quad \text{and} \quad \dot{\hat{t}}_\tau + \|\dot{\hat{\beta}}_\tau\| \leq 1 \tag{25}$$

**Heteroclinic orbit.** *In addition, when $\hat{\beta}_\tau$ is such that $|\hat{\beta}_\tau| \odot \nabla L(\hat{\beta}_\tau) \neq 0$, we have*

$$\dot{\hat{\beta}}_\tau = -\frac{|\hat{\beta}_\tau| \odot \nabla L(\hat{\beta}_\tau)}{\||\hat{\beta}_\tau| \odot \nabla L(\hat{\beta}_\tau)\|} \quad \text{and} \quad \dot{\hat{t}}_\tau = 0. \tag{26}$$

*Furthermore, the loss strictly decreases along the heteroclinic orbits and the path length $\int_0^T \|\dot{\hat{\beta}}_\tau\| \mathrm{d}\tau$ is upperbounded independently of $T$.*

*Proof.* Differentiating Eq. (22) and from the Hessian of $\tilde{\phi}_\alpha$ we get:

$$\begin{aligned} \dot{\hat{\beta}}_\tau^\alpha &= -\dot{\hat{t}}_\tau^\alpha (\nabla^2 \tilde{\phi}_\alpha(\hat{\beta}_\tau^\alpha))^{-1} \nabla L(\hat{\beta}_\tau^\alpha) \\ &= -(1 - \|\dot{\hat{\beta}}_\tau^\alpha\|)(\nabla^2 \tilde{\phi}_\alpha(\hat{\beta}_\tau^\alpha))^{-1} \nabla L(\hat{\beta}_\tau^\alpha). \end{aligned}$$

Therefore taking the norm on the right hand side we obtain that

$$\|\dot{\hat{\beta}}_\tau^\alpha\| = \frac{\|(\nabla^2\tilde{\phi}_\alpha(\hat{\beta}_\tau^\alpha))^{-1}\nabla L(\hat{\beta}_\tau^\alpha)\|}{1 + \|(\nabla^2\tilde{\phi}_\alpha(\hat{\beta}_\tau^\alpha))^{-1}\nabla L(\hat{\beta}_\tau^\alpha)\|},$$

and therefore

$$\dot{\hat{\beta}}_\tau^\alpha = -\frac{(\nabla^2\tilde{\phi}_\alpha(\hat{\beta}_\tau^\alpha))^{-1}\nabla L(\hat{\beta}_\tau^\alpha)}{1 + \|(\nabla^2\tilde{\phi}_\alpha(\hat{\beta}_\tau^\alpha))^{-1}\nabla L(\hat{\beta}_\tau^\alpha)\|}. \tag{27}$$

**Subsequence extraction.** By construction Eq. (22) we have $\dot{\hat{t}}_\tau^\alpha + \|\dot{\hat{\beta}}_\tau^\alpha\| = 1$ , therefore the sequences $(\hat{t}^\alpha)_\alpha$, $(\hat{\beta}^\alpha)_\alpha$ as well as $(\dot{\hat{t}}^\alpha)_\alpha$, $(\dot{\hat{\beta}}^\alpha)_\alpha$ are uniformly bounded on $[0,T]$. The Arzelà-Ascoli theorem yields that, up to a subsequence, there exists $(\hat{t},\hat{\beta})$ such that $(\hat{t}^{\alpha_k},\hat{\beta}^{\alpha_k}) \to (\hat{t},\hat{\beta})$ in $(C^0([0,T],\mathbb{R}\times\mathbb{R}^d), \|\cdot\|_\infty)$. Since $\|\dot{\hat{\beta}}_\tau^\alpha\|, \|\dot{\hat{t}}_\tau^\alpha\| \leq 1$ we have, applying the Banach–Alaoglu theorem, that up to a new subsequence

$$(\dot{\hat{t}}^{\alpha_k},\dot{\hat{\beta}}^{\alpha_k}) \overset{*}{\rightharpoonup} (\dot{\hat{t}},\dot{\hat{\beta}}) \text{ in } L_\infty(0,T) \tag{28}$$

and $\|\dot{\hat{\beta}}_\tau\| \leq \liminf_{\alpha_k}\|\dot{\hat{\beta}}_\tau^{\alpha_k}\| \leq 1$ and thus $\dot{\hat{t}}_\tau + \|\dot{\hat{\beta}}_\tau\| \leq 1$:

$$\int_0^T \|\dot{\hat{\beta}}_\tau\|\mathrm{d}\tau \leq \int_0^T \liminf_{\alpha_k}\|\dot{\hat{\beta}}_\tau^{\alpha_k}\|\mathrm{d}\tau \leq \int_0^{+\infty} \liminf_{\alpha_k}\|\dot{\hat{\beta}}_\tau^{\alpha_k}\|\mathrm{d}\tau \leq \liminf_{\alpha_k}\int_0^{+\infty} \|\dot{\hat{\beta}}_\tau^{\alpha_k}\|\mathrm{d}\tau < C,$$

where the third inequality is by Fatou's lemma. Note that since $[0,T]$ is bounded then it also implies the weak convergence in any $L_p(0,T)$, $1 \leq p < \infty$. Since $(\hat{\beta}^\alpha)$ converges uniformly on $[0,T]$, and $\nabla L$ is continuous, we have that $\nabla L(\hat{\beta}^\alpha)$ converges uniformly to $\nabla L(\hat{\beta})$. Since $\dot{\hat{t}}^{\alpha_k} \rightharpoonup \dot{\hat{t}}$ in $L_1(0,T)$, passing to the limit in the equation $\nabla\tilde{\phi}_\alpha(\hat{\beta}_\tau^\alpha) = -\int_0^\tau \dot{\hat{t}}_s^\alpha\nabla L(\hat{\beta}_s^\alpha)\mathrm{d}s$ leads to

$$-\int_0^\tau \dot{\hat{t}}_s\nabla L(\hat{\beta}_s)\mathrm{d}s \in \partial\|\hat{\beta}_\tau\|_1,$$

due to Lemma 2.

Recall from Eq. (27) and the definition of $\tilde{\phi}_\alpha$ that:

$$\dot{\hat{\beta}}_\tau^\alpha = -\frac{\sqrt{\hat{\beta}_\tau^\alpha + \alpha^4} \odot \nabla L(\hat{\beta}_\tau^\alpha)}{1/\ln(1/\alpha) + \|\sqrt{\hat{\beta}_\tau^\alpha + \alpha^4} \odot \nabla L(\hat{\beta}_\tau^\alpha)\|}. \tag{29}$$

Hence assuming that $\hat{\beta}_\tau$ is such that $\||\hat{\beta}_\tau|\odot\nabla L(\hat{\beta}_\tau)\| \neq 0$, we can ensure that $\||\hat{\beta}_{\tau'}|\odot\nabla L(\hat{\beta}_{\tau'})\| \neq 0$ for $\tau' \in [\tau, \tau+\varepsilon]$ and $\varepsilon$ small enough. We have then $\frac{\sqrt{\hat{\beta}_{\tau'}^\alpha + \alpha^4}\odot\nabla L(\hat{\beta}_{\tau'}^\alpha)}{1/\ln(1/\alpha) + \|\sqrt{\hat{\beta}_{\tau'}^\alpha + \alpha^4}\odot\nabla L(\hat{\beta}_{\tau'}^\alpha)\|}$ converges uniformly toward $-\frac{|\hat{\beta}_{\tau'}|\odot\nabla L(\hat{\beta}_{\tau'})}{\||\hat{\beta}_{\tau'}|\odot\nabla L(\hat{\beta}_{\tau'})\|}$ on $[\tau, \tau+\varepsilon]$. Using the dominated convergence theorem, we have $\int_\tau^{\tau+\varepsilon} \frac{\sqrt{\hat{\beta}_{\tau'}^\alpha + \alpha^4}\odot\nabla L(\hat{\beta}_{\tau'}^\alpha)}{1/\log(1/\alpha) + \|\sqrt{\hat{\beta}_{\tau'}^\alpha + \alpha^4}\odot\nabla L(\hat{\beta}_{\tau'}^\alpha)\|}\mathrm{d}\tau' \to \int_\tau^{\tau+\varepsilon} \frac{|\hat{\beta}_{\tau'}|\odot\nabla L(\hat{\beta}_{\tau'})}{\||\hat{\beta}_{\tau'}|\odot\nabla L(\hat{\beta}_{\tau'})\|}\mathrm{d}\tau'$. We therefore obtain $\dot{\hat{\beta}}_\tau = -\frac{|\hat{\beta}_\tau|\odot\nabla L(\hat{\beta}_\tau)}{\||\hat{\beta}_\tau|\odot\nabla L(\hat{\beta}_\tau)\|}$ in $L_1[0,T]$. Consequently $\|\dot{\hat{\beta}}_\tau\| = 1$ and $\dot{\hat{t}}_\tau = 0$.

**Proof that the loss stricly decreases along the heteroclinic orbits.**

Assume $\hat{\beta}_\tau$ is such that $|\hat{\beta}_\tau|\odot\nabla L(\hat{\beta}_\tau) \neq 0$, then the flow follows

$$\dot{\hat{\beta}}_\tau = -\frac{|\hat{\beta}_\tau|\odot\nabla L(\hat{\beta}_\tau)}{\||\hat{\beta}_\tau|\odot\nabla L(\hat{\beta}_\tau)\|}$$

Letting $\gamma(\tau) = \frac{1}{\||\hat{\beta}_\tau|\odot\nabla L(\hat{\beta}_\tau)\|}$ we get:

$$\mathrm{d}L(\hat{\beta}_\tau) = -\gamma(\tau)\sum_i |\hat{\beta}_\tau(i)|\odot[\nabla L(\hat{\beta}_\tau)]_i^2\mathrm{d}\tau < 0,$$

because $|\hat{\beta}_\tau|\odot\nabla L(\hat{\beta}_\tau)^2 \neq 0$. $\qquad\square$

Borrowing terminologies from [18], we can distinguish two regimes: when $\dot{\hat{\beta}}_\tau = 0$, the system is *sticked* to the saddle point. When $\dot{\hat{t}}_\tau = 0$ and $\|\dot{\hat{\beta}}_\tau\| = 1$ the system switches to a *viscous slip* which follows the normalised flow Eq. (26). We use the term of *heteroclinic orbit* as in the dynamical systems literature since in the weight space $(u, v)$ it corresponds to a path with links two distinct critical points of the loss $F$. Since $\dot{\hat{t}}_\tau = 0$, this regime happens instantly for the original $t$ time scale (*i.e.* a jump occurs).

From Proposition 8, following the same reasoning as in Section 3, we can show that the rescaled process converges uniformly to a continuous saddle-to-saddle process where the saddles are linked by normalized flows.

**Theorem 3.** *Let $T > 0$. For all subsequences defined in Proposition 8, there exist times $0 = \tau_0' < \tau_1 < \tau_1' < \cdots < \tau_p < \tau_p' < \tau_{p+1} = +\infty$ such that the the iterates $(\hat{\beta}_\tau^{\alpha_k})_\tau$ converge uniformly on $[0, T]$ to the following limit trajectory :*

| | | |
|---|---|---|
| **("Saddle")** | $\hat{\beta}_\tau = \beta_k$ | *for $\tau \in [\tau_k', \tau_{k+1}]$ where $0 \leq k \leq p$* |
| **(Orbit)** | $\dot{\hat{\beta}}_\tau = -\dfrac{\|\hat{\beta}_\tau\| \odot \nabla L(\hat{\beta}_\tau)}{\|\|\hat{\beta}_\tau\| \odot \nabla L(\hat{\beta}_\tau)\|}$ | *for $\tau \in [\tau_{k+1}, \tau_{k+1}']$ where $0 \leq k \leq p-1$* |

*where the saddles $(\beta_0 = 0, \beta_1, \ldots, \beta_p = \beta_{\ell_1}^\star)$ are constructed in Algorithm 1. Also, the loss $(L(\hat{\beta}_\tau))_\tau$ is constant on the saddles and strictly decreasing on the orbits. Finally, independently of the chosen subsequence, for $k \in [p]$ we have $\hat{t}_{\tau_k} = \hat{t}_{\tau_k'} = t_k$ where the times $(t_k)_{k \in [p]}$ are defined through Algorithm 1.*

*Proof.* Some parts of the proof are slightly technical. To simplify the understanding, we make use of auxiliary lemmas which are stated in Appendix F. The overall spirit follows the intuitive ideas given in Section 3 and relies on showing that Eq. (25) can only be satisfied if the iterates visit the saddles from Algorithm 1.

We let $\hat{s}_\tau := -\int_0^\tau \dot{\hat{t}}_s \nabla L(\hat{\beta}_s) \mathrm{d}s$, which is continuous and satisfies $\hat{s}_\tau \in \partial \|\hat{\beta}_\tau\|_1$ from Eq. (25). Let $S = \{\beta \in \mathbb{R}^d, |\beta| \odot \nabla L(\beta) = \mathbf{0}\}$ denote the set of critical points and let $(\beta_k, t_k, s_k)$ be the successive values of $(\beta, t, s)$ which appear in the loops of Algorithm 1.

**We do a proof by induction:** we start by assuming that the iterates are stuck at the saddle $\beta_{k-1}$ at time $\tau \geq \tau_{k-1}'$ where $\hat{t}_{\tau_{k-1}'} = t_{k-1}$ and $\hat{s}_{\tau_{k-1}'} = s_{k-1}$ (recurrence hypothesis), we then show that they can only move at a time $\tau_k$ and follow the normalised flow Eq. (26). We finally show that they must end up "stuck" at the new critical point $\beta_k$, validating the recurrence hypothesis.

*Proof of the jump time $\tau_k$ such that $\hat{t}_{\tau_k} = t_k$ :* we set ourselves at time $\tau \geq \tau_{k-1}'$, stuck at the saddle $\beta_{k-1}$. Let $\tau_k := \sup\{\tau, \hat{t}_\tau \leq t_k\}$, we have that $\tau_k < \infty$ from Lemma 3. Note that by continuity of $\hat{t}_\tau$ it holds that $\hat{t}_{\tau_k} = t_k$. Now notice that $\hat{s}_\tau = \hat{s}_{\tau_{k-1}'} - (\hat{t}_\tau - \hat{t}_{\tau_{k-1}'}) \nabla L(\beta_{k-1}) = s_{k-1} - (\hat{t}_\tau - t_{k-1}) \nabla L(\beta_{k-1})$. We argue that for any $\varepsilon > 0$, we cannot have $\hat{\beta}_\tau = \beta_{k-1}$ on $(\tau_k, \tau_k + \varepsilon)$. Indeed by the definition of $\tau_k$ and from the algorithmic construction of time $t_k$, it would lead to $|\hat{s}_\tau(i)| > 1$ for some coordinate $i \in [d]$, which contradicts Eq. (25). Therefore the iterates must move at the time $\tau_k$.

*Heterocline leaving $\beta_{k-1}$ for $\tau \in [\tau_k, \tau_k']$ :* contrary to before, our time rescaling enables to capture what happens during the "jump". We have shown that for any $\varepsilon$, there exists $\tau_\varepsilon \in (\tau_k, \tau_k + \varepsilon)$, such that $\hat{\beta}_{\tau_\varepsilon} \neq \beta_{k-1}$. From Lemma 4, since the saddles are distinct along the flow, we must have that $\hat{\beta}_{\tau_\varepsilon} \notin S$ for $\varepsilon$ small enough. The iterates therefore follow a heterocline flow leaving $\beta_{k-1}$ with a speed of 1 given by Eq. (26). We now define $\tau_k' := \inf\{\tau > \tau_k, \exists \varepsilon_0 > 0, \forall \varepsilon \in [0, \varepsilon_0], \hat{\beta}_{\tau+\varepsilon} \in S\}$ which corresponds to the time at which the iterates reach a new critical point and stay there for at least a small time $\varepsilon_0$. We have just shown that $\tau_k' > \tau_k$. Now from Proposition 8, the path length of $\hat{\beta}$ is finite, and from Lemma 4 the flow visits a finite number of distinct saddles at a speed of 1. These two arguments put together, we get that $\tau_k' < +\infty$ and also $\hat{\beta}_{\tau_k' + \varepsilon} = \hat{\beta}_{\tau_k'}, \forall \varepsilon \in [0, \varepsilon_0]$. On another note, since $\dot{\hat{t}}_\tau = 0$ for $\tau \in [\tau_k, \tau_k']$ we have $\hat{t}_{\tau_k'} = \hat{t}_{\tau_k} (= t_k)$ as well as $\hat{s}_{\tau_k} = \hat{s}_{\tau_k'} (= s_k)$.

*Proof of the landing point* $\beta_k$ : we now want to find to which saddle $\hat{\beta}_{\tau'_k} \in S$ the iterates have moved to. To that end, we consider the following sets which also appear in Algorithm 1:

$$I_{\pm,k} := \{i \in \{1,\ldots,d\}, \text{ s.t. } \hat{s}_{\tau'_k}(i) = \pm 1\} \quad \text{and} \quad I_k = I_{+,k} \cup I_{-,k}. \tag{30}$$

The set $I_k$ corresponds to the coordinates of $\hat{\beta}_{\tau'_k}$ which "are allowed" (but not obliged) to be activated (*i.e.* non-zero). For $\tau \in [\tau'_k, \tau'_k + \varepsilon_0]$ we have that $\hat{s}_\tau = \hat{s}_{\tau'_k} - (\hat{t}_\tau - t_k)\nabla L(\hat{\beta}_{\tau'_k})$. By continuity of $\hat{s}$ and the fact that $\hat{s}_\tau \in \partial\|\hat{\beta}_{\tau'_k}\|_1$, the equality translates into:

- if $i \notin I_k$, $\hat{\beta}_{\tau'_k}(i) = 0$

- if $i \in I_{+,k}$, then $[\nabla L(\hat{\beta}_{\tau'_k})]_i \geq 0$ and $\hat{\beta}_{\tau'_k}(i) \geq 0$

- if $i \in I_{-,k}$, then $[\nabla L(\hat{\beta}_{\tau'_k})]_i \leq 0$ and $\hat{\beta}_{\tau'_k}(i) \leq 0$

- for $i \in I_k$, if $\hat{\beta}_{\tau'_k}(i) \neq 0$, then $[\nabla L(\hat{\beta}_{\tau'_k})]_i = 0$

One can then notice that these conditions exactly correspond to the optimality conditions of the following constrained minimisation problem:

$$\underset{\substack{\beta_i \geq 0, i \in I_{k,+} \\ \beta_i \leq 0, i \in I_{k,-} \\ \beta_i = 0, i \notin I_k}}{\arg\min} \quad L(\beta). \tag{31}$$

We showed in Proposition 7 that the solution to this problem is unique and equal to $\beta_k$ from Algorithm 1. Therefore $\hat{\beta}_\tau = \beta_k$ for $\tau \in [\tau'_k, \tau'_k + \varepsilon_0]$. It finally remains to show that $\hat{\beta}_\tau = \beta_k$ while $\tau \leq \tau_{k+1}$, where $\tau_{k+1} := \sup\{\tau, \hat{t}_\tau = t_{k+1}\}$. For this let $\tau \in [\tau'_k, \tau_{k+1}]$, notice that for $i \notin I_k$, we necessarily have that $\hat{\beta}_\tau(i) = \beta_k(i) = 0$, otherwise we break the continuity of $\hat{s}_\tau$. Similarly, for $i \in I_{k,+}$, we necessarily have that $\hat{\beta}_\tau(i) \geq 0$ and for $i \in I_{k,-}, \hat{\beta}_\tau(i) \leq 0$ for the same continuity reasons. Now assume that $\hat{\beta}_\tau(I_k) \neq \beta_k(I_k)$. Then from Lemma 4 and continuity of the flow, $\exists \tau' \in (\tau'_k, \tau)$ such that $\hat{\beta}_{\tau'} \notin S$ and there must exist a heterocline flow Eq. (26) starting from $\beta_k$ which passes through $\beta_{\tau'}$. This is absurd since along this flow the loss strictly decreases, which is in contradiction with the definition of $\beta_k$ which minimises the problem Eq. (31). $\qquad\square$

## E.1 Proof of Theorem 2

Theorem 3 enables to prove without difficulty Theorem 2 which we recall below. Indeed we can show that any extracted limit $\hat{\beta}$ maps back to the unique discontinuous process $\tilde{\beta}^\circ$.

**Theorem 2.** *Let the saddles* $(\beta_0 = \mathbf{0}, \beta_1, \ldots, \beta_{p-1}, \beta_p = \beta^\star_{\ell_1})$ *and jump times* $(t_0 = 0, t_1, \ldots, t_p)$ *be the outputs of Algorithm 1 and let* $(\tilde{\beta}^\circ_t)_t$ *be the piecewise constant process defined as follows:*

*(Saddles)* $\qquad \tilde{\beta}^\circ_t = \beta_k \qquad\qquad$ *for* $t \in (t_k, t_{k+1})$ *and* $0 \leq k \leq p, \; t_{p+1} = +\infty$.

*The accelerated flow* $(\tilde{\beta}^\alpha_t)_t$ *defined in Eq.* (9) *uniformly converges towards the limiting process* $(\tilde{\beta}^\circ_t)_t$ *on any compact subset of* $\mathbb{R}_{\geq 0}\backslash\{t_1, \ldots, t_p\}$.

*Proof.* We directly apply Theorem 3, let $\alpha_k$ be the subsequence from the theorem. Let $\varepsilon > 0$, for simplicity we prove the result on $[t_1 + \varepsilon, t_2 - \varepsilon]$, all the other compacts easily follow the same line of proof. Note that since $\hat{t}^{\alpha_k}(\tau'_1) \to t_1$ and $\hat{t}^{\alpha_k}(\tau_2) \to t_2$, for $\alpha_k$ small enough $\hat{t}^{\alpha_k}(\tau'_1) \leq t_1 + \varepsilon$ and $\hat{t}^{\alpha_k}(\tau_2) \geq t_2 - \varepsilon$, by the monotonicity of $\tau^{\alpha_k}$, this means that for $\alpha_k$ small enough, $\tau'_1 \leq \tau^{\alpha_k}(t_1 + \varepsilon)$ and $\tau_2 \geq \tau^{\alpha_k}(t_2 - \varepsilon)$. Therefore

$$\sup_{t \in [t_1+\varepsilon, t_2-\varepsilon]} \|\tilde{\beta}^{\alpha_k}_t - \beta_1\| = \sup_{t \in [t_1+\varepsilon, t_2-\varepsilon]} \|\hat{\beta}^{\alpha_k}(\tau_{\alpha_k}(t)) - \beta_1\|$$

$$= \sup_{\tau \in [\tau^{\alpha_k}(t_1+\varepsilon), \tau^{\alpha_k}(t_2-\varepsilon)]} \|\hat{\beta}^{\alpha_k}(\tau) - \beta_1\|$$

$$\leq \sup_{\tau \in [\tau'_1, \tau_2]} \|\hat{\beta}^{\alpha_k}(\tau) - \beta_1\|,$$

which goes uniformly to 0 following Theorem 3. Since this result is independent of the subsequence $\alpha_k$, we get the result of Theorem 2. $\qquad\square$

## E.2 Proof of Proposition 3

We restate and prove Proposition 3 below.

**Proposition 3.** *For all $T > t_p$, the graph of the iterates $(\tilde{\beta}_t^\alpha)_{t \leq T}$ converges to that of $(\hat{\beta}_\tau)_\tau$ :*

$$\text{dist}(\{\tilde{\beta}_t^\alpha\}_{t \leq T}, \{\hat{\beta}_\tau\}_{\tau \geq 0}) \xrightarrow[\alpha \to 0]{} 0 \qquad \text{(Hausdorff distance)}$$

*Proof.* For $\alpha$ small enough, we have that $\hat{t}_{\tau_p'}^\alpha \leq t_p + \varepsilon \leq T$

$$\sup_{\tau \geq 0} d(\hat{\beta}_\tau, \{\tilde{\beta}_t^\alpha\}_{t \leq T}) = \sup_{\tau \leq \tau_p'} d(\hat{\beta}_\tau, \{\tilde{\beta}_t^\alpha\}_{t \leq T})$$

$$\leq \sup_{\tau \leq \tau_p'} \|\hat{\beta}_\tau - \tilde{\beta}_{\hat{t}_\tau^\alpha}^\alpha\|$$

$$= \sup_{\tau \leq \tau_p'} \|\hat{\beta}_\tau - \hat{\beta}_\tau^\alpha\| \xrightarrow[\alpha \to 0]{} 0,$$

according to Theorem 3.

Similarly:

$$\sup_{t \leq T} d(\tilde{\beta}_t^\alpha, \{\hat{\beta}_{\tau'}\}_{\tau'}) = \sup_{\tau \leq \tau_T^\alpha} d(\hat{\beta}_\tau^\alpha, \{\hat{\beta}_{\tau'}\}_{\tau'})$$

$$\leq \sup_{\tau \leq \tau_T^\alpha} \|\hat{\beta}_\tau^\alpha - \hat{\beta}_\tau\| \xrightarrow[\alpha \to 0]{} 0,$$

according to Theorem 3, which concludes the proof. $\square$

# F Technical lemmas

The following lemma describes the behaviour of $\nabla \tilde{\phi}_\alpha(\beta^\alpha)$ as $\alpha \to 0$ in function of the subdifferential $\partial \|\cdot\|_1$.

**Lemma 2.** *Let $(\beta^\alpha)_{\alpha>0}$ such that $\beta^\alpha \xrightarrow[\alpha \to 0]{} \beta \in \mathbb{R}^d$.*

- *if $\beta_i > 0$ then $[\nabla \tilde{\phi}_\alpha(\beta^\alpha)]_i$ converges to $1$*

- *if $\beta_i < 0$ then $[\nabla \tilde{\phi}_\alpha(\beta^\alpha)]_i$ converges to $-1$.*

*Moreover if we assume that $\nabla \tilde{\phi}_\alpha(\beta^\alpha)$ converges to $\eta \in \mathbb{R}^d$, we have that:*

- $\eta_i \in (-1, 1) \Rightarrow \beta_i = 0$

- $\beta_i = 0 \Rightarrow \eta_i \in [-1, 1]$.

*Overall, assuming that $(\beta^\alpha, \nabla \tilde{\phi}_\alpha(\beta^\alpha)) \xrightarrow[\alpha \to 0]{} (\beta, \eta)$, we can write:*

$$\eta \in \partial \|\beta\|_1.$$

*Proof.* We have that

$$[\nabla \tilde{\phi}_\alpha(\beta^\alpha)]_i = \frac{1}{2 \ln(1/\alpha)} \operatorname{arcsinh}(\frac{\beta_i^\alpha}{\alpha^2})$$

$$= \frac{1}{2 \ln(1/\alpha)} \ln \Big( \frac{\beta_i^\alpha}{\alpha^2} + \sqrt{\frac{(\beta_i^\alpha)^2}{\alpha^4} + 1} \Big).$$

Now assume that $\beta_i^\alpha \to \beta_i > 0$, then $[\nabla \tilde{\phi}_\alpha(\beta^\alpha)]_i \to 1$, if $\beta_i < 0$ we conclude using that $\operatorname{arcsinh}$ is an odd function. All the claims are simple consequences of this. $\square$

The following lemma shows that the extracted limits $\hat{t}$ as defined in Proposition 8 diverge to $\infty$. This divergence is crucial as it implies that the rescaled iterates $(\hat{\beta}_\tau)_\tau$ explore the whole trajectory..

**Lemma 3.** *For any extracted limit $\hat{t}$ as defined in Proposition 8, we have that $\tau - C \leq \hat{t}_\tau$ where $C$ is the upperbound on the length of the curves defined in proposition 6.*

*Proof.* Recall that

$$\tau^\alpha(t) = t + \int_0^t \|\dot{\tilde{\beta}}_s^\alpha\| \mathrm{d}s.$$

From Proposition 6, the full path length $\int_0^{+\infty} \|\dot{\beta}_s^\alpha\| \mathrm{d}s$ is finite and bounded by some constant $C$ independently of $\alpha$. Therefore $\tau^\alpha$ is a bijection in $\mathbb{R}_{\geq 0}$ and we defined $\hat{t}_\tau^\alpha = (\tau^\alpha)^{-1}(\tau)$. Furthermore $\tau^\alpha(t) \leq t + C$ leads to $t \leq \hat{t}^\alpha(t + C)$ and therefore $\tau - C \leq \hat{t}^\alpha(\tau)$ for all $\tau \geq 0$. This inequality still holds for any converging subsequence, which proves the result. $\square$

Under a mild additional assumption on the data (see Assumption 2), we showed after the proof of Proposition 1 in Appendix B that the number of saddles of $F$ is finite. Without this assumption, the number of saddles is *a priori* not finite. However the following lemma shows that along the flow of $\hat{\beta}$ the number of saddles which can potentially be visited is indeed finite.

**Lemma 4.** *The limiting flow $\hat{\beta}$ as defined in Proposition 8 can only visit a finite number of critical points $\beta \in S := \{\beta \in \mathbb{R}^d, \beta \odot \nabla L(\beta) = \mathbf{0}\}$ and can visit each one of them at most once.*

*Proof.* Let $\tau \geq 0$, and assume that $\hat{\beta}_\tau \in S$, i.e., we are at a critical point at time $\tau$. From Proposition 1, we have that

$$\hat{\beta}_\tau \in \underset{\beta_i = 0 \text{ for } i \notin \operatorname{supp}(\hat{\beta}_\tau)}{\arg \min} L(\beta), \tag{32}$$

Let us define the sets

$$I_{\pm} := \{i \in \{1, \ldots, d\}, \text{ s.t. } \hat{s}_\tau(i) = \pm 1\} \quad \text{and} \quad I = I_+ \cup I_-.$$

The set $I$ corresponds to the coordinate of $\hat{\beta}_\tau$ which "are allowed" (but not obliged) to be non-zero since from Eq. (25), $\text{supp}(\hat{\beta}_\tau) \subset I$. Now given the fact that the sub-matrix $X_I = (\tilde{x}_i)_{i \in I} \in \mathbb{R}^{n \times \text{card}(I)}$ is full rank (see part (1) of the proof of Proposition 7 for the explanation), the solution of the minimisation problem (32) is unique and equal to $\beta[\xi] = (X_\xi^\top X_\xi)^{-1} X_\xi^\top y$ and $\beta[\xi^C] = 0$ where $\xi = \text{supp}(\hat{\beta}_\tau)$. There are $2^d = \text{Card}(P([d]))$ (where $P([d])$ contains all the subsets of $[d]$) number of constraints of the form $\{\beta_i = 0, i \notin \mathcal{A}\}$, where $\mathcal{A} \subset [d]$, and $\hat{\beta}_\tau$ is the unique solution of one of them. $\hat{\beta}_\tau$ can therefore take at most $2^d$ values (very crude upperbound). There is therefore a finite number of critical points which can be reached by the flow $\hat{\beta}$. Furthermore, from Proposition 8, the loss is strictly decreasing along the heteroclinic orbits, each of these critical points can therefore be visited at most once. $\qquad \square$

