# OpenReview forum: "Saddle-to-Saddle Dynamics in Diagonal Linear Networks"
_NeurIPS.cc/2023/Conference — NeurIPS 2023 spotlight_

### Official Review · Reviewer_tpj2 · 2023-06-22

**Soundness:** 3 good
**Presentation:** 3 good
**Contribution:** 3 good
**Rating:** 7
**Confidence:** 3

**Summary:**

This paper characterizes the trajectory of gradient flow on 2-layer diagonal linear networks for linear regression tasks. Specifically, the paper considers the model parameterized as $x \mapsto \langle u \odot w, x \rangle$ in the linear regression setting. By interpreting the gradient flow on the nonconvex loss as a mirror flow, the authors show, in the limit of vanishing initialization, that the gradient flow dynamics jump between various saddles before converging to the minimum $\ell_1$ norm solution. The paper characterizes the exact times of these jumps (after appropriate rescaling based on the initialization size $\alpha$ in the $\alpha \rightarrow 0$ limit), as well as the location of these saddles; these times and locations can be computed via algorithm 1. As a consequence, the paper shows that each saddle is a solution to a constrained minimization problem where some subset of coordinates are fixed 0. Under a RIP assumption on the data, the paper shows that new coordinates are added sequentially at each saddle, thus demonstrating incremental learning.

**Strengths:**

- The paper appears to be technically sound, and is well-written and easy to understand.
- The quadratically parameterized regression problem has been well studied in prior works (Woodworth et al. 2020, Azulay et al. 2021, and others), which show that the gradient flow can be characterized by mirror descent and converges to the minimum $\ell_1$ solution in the $\alpha \rightarrow 0$ limit. However, the current paper exactly characterizes the limit of the gradient flow trajectory under minimal data assumptions, which is a novel analysis. The main technical novelty here is proving that as $\alpha \rightarrow 0$ the trajectory converges to a piecewise constant process.
- While many prior works on implicit regularization only focus on properties of the solution at convergence, the current paper can describe the saddle points along the trajectory. I find it to be a significant contribution that the successive saddles which can be visited by gradient flow can be characterized (by Algorithm 1) explicitly. In particular, the observation that coordinates can be deactivated has not appeared in prior work and is quite interesting.
- Finally, I find this paper to be of moderate significance, as the quadratically parameterized linear regression setting is a common toy problem to understand the implicit regularization effect of gradient descent more generally.

**Weaknesses:**

There are a couple (minor) weaknesses of the current theory.
- The analysis is limited to the quadratically parameterized regression setting, which while despite possessing rich implicit regularization behavior is still far from more practical settings in which implicit regularization occurs.
- One weakness of the current paper is that the jump times and saddle locations are only implicitly defined via Algorithm 1. The paper does show that under an RIP assumption that the jump times are $1/\beta^*_s$ and the saddles correspond to incrementally learning more coordinates. However, it is difficult to interpret the intermediate iterates of Algorithm 1 more generally, which reduces the impact of this paper.
- Another weakness is that the entirety of the analysis is done in the $\alpha \rightarrow 0$ limit, rather than at some small (but finite) initialization $\alpha$.

Minor typos:
- line 133 “diferred” → “deferred”
- line 318 “independant” → independent”

**Questions:**

- In general, what can one say about successive saddles? It would be interesting to understand more fine grained properties about the sequence of saddles, such as how many coordinates can change at each step and under what conditions coordinates can get deactivated. I find it unlikely that Algorithm 1 would loop through all possible subsets of activated coordinates, and instead find it more likely that coordinates would be deactivated less frequently. I thus think the paper would benefit from additional investigation (either theoretically or empirically) into how the set of activated coordinates changes between saddles.
- What can be said in the case when $\alpha$ is small but isn’t taken to 0? Can one say something quantitatively about whether saddle-to-saddle dynamics occur, or the rate at which the trajectory limits to the piecewise constant trajectory $\tilde \beta^\circ$?

**Limitations:**

Limitations/Broader Impact are adequately addressed.

---

> ### Author Rebuttal · Authors · 2023-08-09
>
> Thank you very much for the feedback.
>
> We answer to your questions below and answer to your comment on the restrictive setting in the *official comment* section since it was made by nearly all the reviewers.
>
> **Finite initialisation**: unfortunately, as we acknowledge line 267, our results do not provide a rate of convergence in $\alpha$ and therefore cannot explain the observed stepwise trajectories for a non-zero initialization. We believe that a non-asymptotic result is currently challenging, but we believe that our work is nonetheless a significant contribution and that the tools and techniques we provide could eventually lead to a non-asymptotic result.
>
> **Additional results concerning Algorithm 1**: we would first start by highlighting that the main goal of our work is to prove that the limiting dynamics of GF for vanishing initalisation is a saddle-to-saddle dynamics which is fully described by Algorithm 1. From there, as rightfully mentioned by the reviewer, many very interesting questions can be raised concerning the behaviour of this algorithm. We already provide a few:
> - upper bound on the number of iterations
> - proof that the  coordinates of the iterates $\beta_k$ have at most $n$ non-zero coordinates (see prop 6 in the appendix)
> - analysis of the full behaviour under a RIP assumption
>
> Moreover, we believe that further results are out of the scope of our paper. Indeed, to draw a comparison, we are not aware of results for the Homotopy method which go beyond results under RIP assumptions apart from [a] which provides a worst-case lower-bound on the number of iterations. However, note that this result is from 2012, therefore more than 10 years after that the algorithm was initially introduced in [b].
>
> [a] Mairal and Yu, Complexity Analysis of the Lasso Regularization Path, ICML 2012
>
> [b] Osborne et al. A new approach to variable selection in least squares problems, 2000.

---

> > ### Comment · Reviewer_tpj2 · 2023-08-11
> > **Response to Authors**
> >
> > Thank you to the authors for responding to my questions. I agree with the authors that the current results are indeed novel and relevant, and that the additional results I asked about are beyond the scope of the current paper (though still interesting!). I have increased my score from a 6 to a 7.

---

### Official Review · Reviewer_P2pV · 2023-07-04

**Soundness:** 4 excellent
**Presentation:** 4 excellent
**Contribution:** 2 fair
**Rating:** 8
**Confidence:** 5

**Summary:**

This work concerns the behavior of gradient flow of 2-layer diagonal linear neural networks when the initialization scale goes to zero. The authors show that this limiting behavior is governed by a piecewise constant trajectory consisting of jumps from saddle to saddle.

**Strengths:**

The incremental learning and saddle-to-saddle phenomenon is interesting. Although the model considered in this paper is equivalent to linear models, the dynamics that the authors uncover is nonetheless nontrivial and of broader interest. The writing quality is excellent and the technical material is presented in an intuitive to understand way.

**Weaknesses:**

Is the final point $\beta_p$ in Theorem 2 the minimum $\ell_1$ solution to $L$? Theorem 2 does not explicitly state it.

Much of the later analysis is regarding the $\beta$'s. But the model is stated in terms of the $u$'s and $v$'s. Can the authors comment on the behavior of the $u$'s and $v$'s? (Or at least point me to where this was discussed, in case I missed it).

line 40 "diagonal linear networks which are simplified neural networks that have received significant attention lately...". Could the authors provide a slightly expanded discussion of what these prior work did? Since this paper is specifically about diagonal linear networks, this part of the related work I believe is especially important.

**Questions:**

What is the significance of the connection to the Homotopy algorithm? I'm not familiar with it and would appreciate if the authors can explain a bit more why this connection matters.

If the authors can address this and the weaknesses above, I'll be happy to raise the score to a 7.

**Limitations:**

I do not believe the authors addressed the limitations, at least not in an explicit way. I think the paper would be greatly improved if it can touch upon the comment brought up on line 78 that the activation function is the identity. The authors make it very clear that the work is about linear networks, as many other prior works have done. I feel like if the authors can provide some form of experimental results/explanation of whether this phenomenon appear (or does not appear) where $\sigma$ is nonlinear would be greatly beneficial. That said, I think the paper's technical contribution outweighs this.

If the authors can address this and all of the above, I'll be happy to raise the score to an 8.

---

> ### Author Rebuttal · Authors · 2023-08-09
>
> Thank you very much for the extensive and valuable feedback. We answer to your remarks and questions below.
>
> **Final point $\beta_p$**: the final point $\beta_p$ is indeed the minimum $\ell_1$ norm solution $\beta^*_{\ell_1}$, we tried to make this clear by pointing out that $\beta_p = \beta^*_{\ell_1}$ in Theorem 2 line 253. We propose to emphasize this point if the reviewer feels it is necessary.
>
> **Neuron point of view**: we indeed did not put much emphasis on the neurons $(u,v)$. However as explained lines 263-265, there exists a bijection (given in Lemma 1) between the neurons and the vectors $\beta_t$. Therefore we immediately get that the neurons $(u^\alpha_t, v^\alpha_t)$ converge towards $(\sqrt{\vert \tilde{\beta}_t^\circ \vert}, \text{sign}(\tilde{\beta}_t^\circ)  \sqrt{\vert \tilde{\beta}^\circ_t \vert})$.  We will make this point clearer in the revised version.
>
> **Previous works on DLNs**: by lack of space we indeed unfortunately did not provide an extensive discussion on previous works on DLNs. This omission is mostly because the vast majority of these works investigate the properties of the solution recovered by gradient flow / GD / SGD, without looking at the trajectory. The two most relevant references are [4] and [38] and are discussed at the beginning of section 2.2:
> - [4] shows that the iterates $\beta_t^\alpha$ follow a mirror flow with potential $\phi_\alpha$
> - [38] shows that the recovered solution $\beta_\infty^\alpha$ minimises the potential $\phi_\alpha$
>
> Concerning other works on diagonal linear networks, we have:
> - [33] investigates the role of SGD’s noise on the recovered solution
> - [18] investigates the effect of label noise on the recovered solution
> - [13] and [c] investigate the effect of the stepsize on (S)GD's recovered solution
> - [37] investigates the statistical properties of the solution recovered by GD under a RIP assumption
>
> We will add the discussion concerning these results in the related work.
>
> **Link with the Homotopy algorithm**: the connection between Algorithm 1 and the Homotopy algorithm **is not important in itself but should be seen as a help to understand Alg 1**.
> We mention this connection for two reasons: *(1)* the main one is to make Algorithm 1 feel less opaque and "out of the blue" by linking it to an old and well-established algorithm, *(2)* the construction of Algorithm 1 as given section 3.1 follows very similar lines as that of the Homopy algorithm.
> In this way, making a link with the Homotopy algorithm is a way (for readers who are acquainted to it) of providing a familiar algorithm which makes our results more natural. **However, we hope that our intuitive construction of Alg 1 in Section 3 is clear enough for the algorithm to be understandable to people who are not familiar with the Homotopy algorithm.**
>
> **Non-linear activation**: We would like to emphasis that **saddle-to-saddle dynamics also occur even when there is a non-linear activation**. To support this claim, we refer to [22] which "give evidence for the hypothesis that, as iterations progress, SGD learns functions of increasing complexity " for non-linear deep-networks and that there is a " separation in phases for learning". Furthermore, in [8], a 2-layer ReLU network is considered and you can observe a clear saddle-to-saddle dynamics in Figure 3 (for orthogonal inputs) and also in Figure 4 (f) (general inputs). We can also mention more recent works which consider non-linear networks and where clear saddle-to-saddle dynamics are observed: see Figure 2 in [a] and Figure 5 in [b]. We hope that these references will convince the reviewer that a non-linear activation function **does not** prevent the occurrence of saddle-to-saddle phenomenons. We will make this clearer in the revised version.
>
> [a] Simon et al., On the Stepwise Nature of Self-Supervised Learning, arXiv 2023
>
> [b] Oswald et al., Transformers Learn In-Context by Gradient Descent, arXiv 2023
>
> [c] Nacson et al., Implicit Bias of the Step Size in Linear Diagonal Neural Networks, ICML 2022

---

> > ### Comment · Reviewer_P2pV · 2023-08-11
> >
> > I thank the authors for a thorough reply. I have raised my score to an 8.

---

### Official Review · Reviewer_8NQr · 2023-07-05

**Soundness:** 3 good
**Presentation:** 3 good
**Contribution:** 3 good
**Rating:** 6
**Confidence:** 3

**Summary:**

This paper analyzes a two-layer diagonal linear network, which is a linear regression model where the linear weight is parameterized as the point-wise product of two weight vectors. It is shown that, with vanishing initialization, gradient flow will jump between saddle points of the training loss and eventually reach the $\ell_1$-minimum-norm solution. This paper further provides an algorithm to calculate all the saddle points.

**Strengths:**

It is very important to understand the training dynamics of neural networks, and this paper analyzes a novel and interesting saddle-to-saddle phenomenon. Moreover, Algorithm 1 in this paper can compute all the jump times and saddle points, giving a complete characterization of the gradient flow trajectory. The writing is also clean, with illustrative examples and graphs.

**Weaknesses:**

The main weakness is that the two-layer diagonal linear network model (specifically, the point-wise product of two weight vectors) is too simple. If there is some evidence that some of form of the saddle-to-saddle phenomenon also happens in practical networks, this paper will be more convincing.

**Questions:**

N/A

---

> ### Author Rebuttal · Authors · 2023-08-09
>
> Thank you very much for your feedback.
>
> We answer to your questions below and answer to your comment on the restrictive setting in the *official comment* section since it was made by nearly all the reviewers.
>
> **Evidence for practical networks**:  we would like to re-emphasize that saddle-to-saddle dynamics **have been observed in practice in various cases** and that this preponderance is precisely the motivation behind our work (see lines 25 to 32 in our introduction).  For instance, in [22], the authors empirically *give evidence for the hypothesis that, as iterations progress, SGD learns functions of increasing complexity* when training deep neural networks.  For further references highlighting the incremental nature of learning, you can see characteristic stepwise learning curves in Figures 1 and 2 from [a] or Figure 2 and 3 from [b]. Also note that these observations in practical networks have led to a large amount of works trying to theoretically prove such dynamics in various settings: matrix factorisation, linear networks etc. (see lines 33 to 41 in our introduction). However, our work is the first to theoretically provide a complete picture describing this phenomenon without imposing restrictive assumptions on the design matrix.
>
> [a] Simon et al., On the Stepwise Nature of Self-Supervised Learning, arXiv 2023
>
> [b] Oswald et al, Transformers Learn In-Context by Gradient Descent, arXiv 2023

---

> > ### Comment · Reviewer_8NQr · 2023-08-18
> >
> > Thanks for the response! I will keep my score.

---

### Official Review · Reviewer_FL6c · 2023-07-07

**Soundness:** 3 good
**Presentation:** 3 good
**Contribution:** 3 good
**Rating:** 7
**Confidence:** 4

**Summary:**

In this paper, the authors studied the training dynamics of gradient flow that minimizes mean-square loss with 2-layer diagonal linear networks and data in general position. Under the small initialization (initialization scale goes to 0), the authors showed that the limiting dynamics follows a saddle-to-saddle dynamics (jump from one saddle point to another saddle point) until reach the min-$\ell_1$-norm interpolator. They gave an algorithm that can compute the visited saddle points. The results generalized the previous works on incremental learning. Experiments are also provided to verify the results.

**Strengths:**

1.	The paper is clearly written and easy-to-follow. The proof sketch and example are given to help the readers to understand the proof easier.
2.	Understanding the training dynamics of neural networks/nonlinear models is an interesting and important problem.
3.	This paper gives a precise characterization of the training dynamics that jumps between saddle points. This recovers and goes beyond some of the previous works on the incremental learning for linear diagonal networks.
4.	The technique of reparametrized time to “accelerate” time seems to be interesting and might be of independent interest.


**Weaknesses:**

1.	The current paper focuses on the linear diagonal linear networks. It would be interesting to see if such analysis could be generalized to other more complicated problems.

**Questions:**

1.	In Theorem 2, it seems that the final solution is min-$\ell_1$-norm interpolator, so I think that means we are implicitly assuming the input dimension $d$ is at least the number of samples $n$?

**Limitations:**

The limitation is discussed in the paper. This is a theoretical work and therefore does not seem to have negative societal impact.

---

> ### Author Rebuttal · Authors · 2023-08-09
>
> Thank you very much for the valuable feedback.
>
> We answer to your comment on the restrictive setting in the *official comment* section since it was made by nearly all the reviewers.
>
> **Dimension and number of samples**: your comment on the input dimension $d$ having to be larger than the number of samples $n$ is due to our imprecise definition of $\beta^*_{\ell_1}$ in the paper, we make it clearer here and will make it clearer in the revised version. The definition of $\beta^*_{\ell_1}$ which we give line 116 as $\beta^*_{\ell_1} = \arg \min_{X \beta = y} \Vert \beta \Vert_1$ indeed **only makes sense when $d > n$**. However, when $d \leq n$, all our results still hold by simply letting $\beta^*_{\ell_1}$ be the unique minimiser of the loss (which trivially is still the minimum l1-norm solution as it is then the only solution!). The general and correct (but a bit more heavy) way of defining $\beta^*_{\ell_1}$ is as :
> $\beta^*_{\ell_1} \coloneqq \arg \min_{ \beta \in \arg \min L} \Vert \beta \Vert_1$. This definition then holds for any $(n, d)$ and all our results still hold. Thank you for pointing out this imprecision.

---

> > ### Comment · Reviewer_FL6c · 2023-08-11
> >
> > Thanks for the response to address my question. I will keep my score.

---

### Official Review · Reviewer_BqoH · 2023-07-10

**Soundness:** 2 fair
**Presentation:** 3 good
**Contribution:** 2 fair
**Rating:** 5
**Confidence:** 3

**Summary:**

This paper studies the saddle-to-saddle dynamics in Diagonal Linear Networks.
The authors present solid theoretical understanding.
They show that over 2-layer diagonal linear network, gradient flow starting with vanishing initialization visits then jump from saddles  jumps from a saddle of the training loss to another until reaching the minimum $\ell_1$-norm solution.



**Strengths:**

The paper is clear and well-written.
Understanding the training dynamics of gradient descent over neural networks is a significant theoretical issue.
Particularly, the phenomenon of saddle-to-saddle during neural network training remains mysterious, and this paper makes a valuable contribution by offering a solid theoretical analysis for 2-layer diagonal linear networks.
The proof presented in the paper incorporates innovative techniques, including mirror flow and time-reparametrization.

**Weaknesses:**

The article's significance may be constrained by the focus on 2-layer diagonal linear networks, which have limited representation abilities and are no better than linear models.
However, this limitation is not a major concern since the problem itself is non-convex, even in this simplified scenario.

**Questions:**

1. I would like to know how much the vanishing initialization would affect the saddle-to-saddle phenomenon. Will saddle-to-saddle dynamics also occur if a practical initialization is used?

2. As shown in the theory and Figure 2, the recovery of coordinates is sequential, which seems to be similar to the Coordinate Descent algorithm. In this setting of 2-layer diagonal linear networks, are GD and CD inherent related?

**Limitations:**

The analysis in this study focus on 2-layer diagonal linear networks, which have limited representation abilities and exhibit some special properties such as Prop 1.

---

> ### Author Rebuttal · Authors · 2023-08-09
>
> Thank you very much for the valuable feedback.
>
> We answer to your questions below and answer to your comment on the restrictive setting in the *official comment* section since it was made by nearly all the reviewers. We answer to both of your questions below.
>
> **Practical initialisations and saddle-to-saddle**: Saddle-to-saddle type of dynamics (which can also be referred to as the feature learning regime in the deep learning community) have been empirically observed in various cases **with practical initialisations**. For instance, in [22], the authors empirically *"give evidence for the hypothesis that, as iterations progress, SGD learns functions of increasing complexity"* when training deep neural networks. They emphasize that these observations are for *"standard random initialisation"*.  For further references highlighting the incremental nature of learning for practical initialization, you can observe characteristic stepwise learning curves in Figures 1 and 2 from [a] as well as Figures 2 and 3 from [b]. Therefore, saddle-to-saddle dynamics **do not** simply correspond to a pathological and uninteresting phenomenon which only occurs asymptotically. They gradually appear and then amplify when taking the initialization to zero. In our setting, you can observe the progressive sharpening of the curves in Figure 3 (left) in the appendix. A very interesting question for future work would be to theoretically exhibit the threshold $\alpha_0$ at which the saddle-to-saddle dynamics starts to appear. However this would require a non-asymptotic and different type of analysis compared to the one we propose in our work.
>
> **Algorithm 1 and CD**: Algorithm 1 and Coordinate Descent share similarities as they both iteratively solve minimisation problems over a fixed set of coordinates. The two are nonetheless distinct algorithms, the major difference is that Algorithm 1 progressively identifies the support of the minimum l1 interpolator: the successive sets of minimized coordinates are appropriately chosen and are of overall increasing size. On the other hand, for CD, only **one coordinate at a time** is updated and this coordinate is chosen very differently to Alg 1: in many cases it is simply chosen randomly. A major consequence of these differences is that Alg 1 terminates in a finite number of iterations while CD only converges asymptotically.
>
> [a] Simon et al., On the Stepwise Nature of Self-Supervised Learning, arXiv 2023
>
> [b] Oswald et al, Transformers Learn In-Context by Gradient Descent, arXiv 2023

---

> > ### Comment · Reviewer_BqoH · 2023-08-11
> >
> > Thanks for the reviewer's detailed response to my questions.

---

### Author Rebuttal · Authors · 2023-08-09

We thank all the reviewers for the time they spent reviewing our paper and for the valuable feedback. An overall comment made by all reviewers (except reviewer P2pV) is that the considered setting (2-layer diagonal linear network) is too restrictive. We naturally agree that our setting is very far from practical networks. However, we want to emphasize that despite its apparent simplicity, the loss function is non-convex and very rich behaviors already occur, as illustrated in the paper.
This saddle-to-saddle behavior (also called incremental learning or feature learning) has been observed in a variety of more complicated settings as motivated in the introduction. We take advantage of this rebuttal to add two recent works which highlight the prominence of such dynamics in current deep learning settings: see Figure 2 in [a] where several network architectures are considered as well as Figures 2, 3, 5 in [b] where various transformers are studied.

As such, we consider our setting as being an ideal proxy model for gaining a deeper understanding of saddle-to-saddle dynamics. We would also like to highlight that despite the apparent simplicity of the model, the analysis is already rather involved and a non-trivial algorithm emerges. Furthermore, we believe that the tools we leverage in our work could be very useful for future works studying similar dynamics but in different settings. Finally, we want to point out that the solution recovered by gradient flow with vanishing initialization is still not fully understood for more complex frameworks such as matrix multiplication or 2-layer ReLU networks. Hence we believe that studying the full trajectory and explaining the observed saddle-to-saddle dynamics in such general settings is currently out of reach.

In view of all this, we hope that the reviewers will acknowledge that though our framework may seem deceptively simple, the analysis is already quite intricate and our work contributes to the understanding of an important topic as well as provides useful tools for future research.

[a] Simon et al., On the Stepwise Nature of Self-Supervised Learning, arXiv 2023

[b] Oswald et al, Transformers Learn In-Context by Gradient Descent, arXiv 2023

---

> ### Comment · Area_Chair_oVW3 · 2023-08-18
> **Thank you for the rebuttal**
>
> Dear authors,
>
> thank you for providing a rebuttal. Most of the reviewers have already replied, so this is just to let you know that I am in contact with the remaining one as well.
>
> Best,
> Your AC

---

### Decision · Program_Chairs · 2023-09-21

**Decision:**

Accept (spotlight)

**Comment:**

This paper considers two-layer diagonal linear networks for a regression problem, and the main result is a precise characterisation of the trajectory of gradient flow. Specifically, a saddle-to-saddle dynamics is unveiled and described accurately.

All reviewers are positive about the work (although with varying levels of enthusiasm). As pointed out in multiple reviews and also acknowledged by the authors in the rebuttal, the setting of two-layer diagonal linear networks is a bit restrictive. Nonetheless, I agree with the authors when they claim that 'very rich behaviours already occur'. The insights shed by the paper are certainly non-trivial, and there is technical novelty as well. For these reasons, after my own reading of the reviews, rebuttal and paper, I am happy to recommend acceptance.